# *TinyMIG*: Transferring Generalization from Vision Foundation Models to Single-Domain Medical Imaging

Chuang Liu [* 1 2]  Hongyan Xu [* 3]  Yichao Cao [3]  Xiu Su [3]  Zhe Qu [3]  Tianfa Li [4]  Shan An [5]  Haogang Zhu [1 2 6]

## Abstract

Medical imaging faces significant challenges in single-domain generalization (SDG) due to the diversity of imaging devices and the variability among data collection centers. To address these challenges, we propose **TinyMIG**, a framework designed to transfer generalization capabilities from vision foundation models to medical imaging SDG. TinyMIG aims to enable lightweight specialized models to mimic the strong generalization capabilities of foundation models in terms of both global feature distribution and local fine-grained details during training. Specifically, for global feature distribution, we propose a Global Distribution Consistency Learning strategy that mimics the prior distributions of the foundation model layer by layer. For local fine-grained details, we further design a Localized Representation Alignment method, which promotes semantic alignment and generalization distillation between the specialized model and the foundation model. These mechanisms collectively enable the specialized model to achieve robust performance in diverse medical imaging scenarios. Extensive experiments on large-scale benchmarks demonstrate that TinyMIG, with extremely low computational cost, significantly outperforms state-of-the-art models, showcasing its superior SDG capabilities. All the code and model weights will be publicly available.

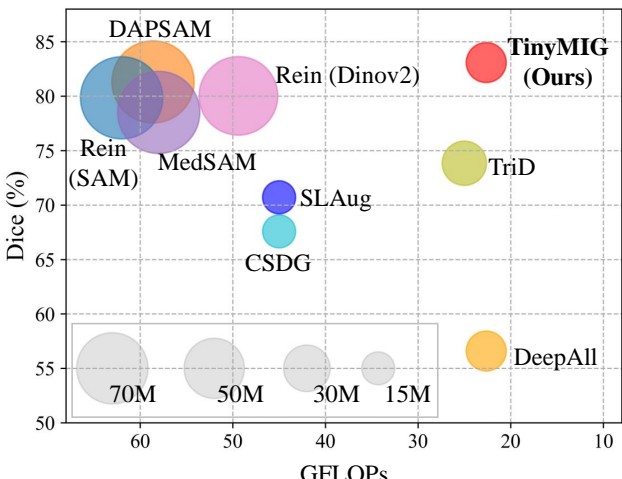

*Figure 1.* The proposed TinyMIG significantly surpasses the SDG capabilities of SOTAs with extremely low computational cost and model parameter size.

## 1. Introduction

In the field of medical imaging analysis, *Domain Adaptation* (DA) (Kouw & Loog, 2021; Guan & Liu, 2021a) and *Domain Generalization* (DG) (Zhou et al., 2022a; Yoon et al., 2023; Huang et al., 2020) address the critical challenge of managing variations in data distributions that deviate from the conventional assumption of independent and identically distributed (IID) data (Carlucci et al., 2019). Among these, *Single-source Domain Generalization* (SDG) tackles the practical problem of developing methods capable of generalizing from a single source to multiple out-of-distribution (OOD) target domains (Liu et al., 2022b; Xu et al., 2023). This task is especially relevant in medical imaging, where data availability is often limited, and privacy requirements are stringent (Price & Cohen, 2019). In recent years, key approaches in SDG have been proposed include Data Augmentation (Liu et al., 2024; Zhou et al., 2021; Su et al., 2022b), Regularization (Chen et al., 2022; Zhang et al., 2022), and Self-supervised Learning (Zhou et al., 2022b), each of which aims to improve model robustness and adaptability to unseen domains. Despite these advancements, significant challenges remain in comprehensively addressing the SDG problem, such as limited generalization capabilities and high computational complexity, highlighting the

---

[*]Equal contribution  [1]State Key Laboratory of Complex & Critical Software Environment, Beihang University, China [2]Zhongguancun Laboratory, China [3]Big Data Institute, Central South University, China [4]Jinan Institute of Supercomputing Technology [5]School of Electrical and Information Engineering, Tianjin University, China [6]Hangzhou International Innovation Institute, Beihang University, China. Correspondence to: Xiu Su <xiusu@csu.edu.cn>, Haogang Zhu <haogangzhu@buaa.edu.cn>.

*Proceedings of the 42$^{nd}$ International Conference on Machine Learning*, Vancouver, Canada. PMLR 267, 2025. Copyright 2025 by the author(s).

ongoing need for further research in this area.

Regarding model generalization capabilities, one of the most notable recent advancements in computer vision (CV) and natural language processing (NLP) has been the development of Foundation Models (FMs) or Large Models (LMs), such as CLIP (Radford et al., 2021), SAM (Kirillov et al., 2023b), and GPT (Brown et al., 2020). In this work, we prefer to use the term Vision Foundation Models **VFMs** to refer to these methods. Leveraging large-scale parameters and extensive pre-training datasets, these models demonstrate superior generalization capabilities across a range of downstream tasks. The latest developments in these areas inspire us to reflect on two key questions: *i*): *Whether the generalization capabilities of VFMs can enhance smaller specialized models, particularly in specialized fields such as medical imaging. ii*): *Additionally, how to design effective knowledge transfer methods to facilitate the improvement of specialized models by leveraging VFMs.*

In this work, we propose **TinyMIG** as shown in Fig. 2, a method to transfer generalization proficiency from VFMs to medical imaging SDG. The core objective of this framework is to enable smaller specialized models to mimic the strong generalization capabilities of large foundation models in terms of both global feature distribution and local fine-grained details during training, so that the specialized models can independently handle domain shifts in unseen target domains during testing. Specifically, for global feature distribution, we propose a feature redistribution consistency learning strategy that mimics the prior distributions of the VFMs layer by layer. For local fine-grained details, to further enhance the specialized model's learning of fine-grained features from the foundation model, we design a localized representation transfer method, which promotes semantic alignment and generalization distillation between the specialized model and the VFMs. TinyMIG framework offers the following advantages: *i*) Computational efficiency: eliminates the need to incur the substantial computational overhead of foundation models during the inference phase; *ii*) Structural flexibility: Seamlessly integrates various foundation models to enhance the performance of domain-specific models; *iii*) Superior performance: achieving exceptional results with minimal parameters and faster inference speeds, as shown in Fig. 1. The contributions of this work are summarized as follows:

- We propose an efficient framework for transferring the generalization capabilities of large-scale Foundation Models to lightweight specialized models. This framework maintains robust SDG in medical imaging while requiring only minimal computational overhead.

- For global feature distribution, we propose a feature distribution consistency learning strategy that encourages the specialized model to hierarchically mimic the prior distributions of the foundation model.

- For local fine-grained details, we design a domain-invariant representation transfer method, which achieves improved localized representation alignment through frequency-domain discriminative content enhancement.

- Extensive experiments on public datasets demonstrate that TinyMIG significantly outperforms existing methods with minimal model size and computational cost.

## 2. Related Work

**Domain Generalization (DG).** DG has been a prominent research focus in the medical imaging field (Wang et al., 2020; Guan & Liu, 2021b; Su et al., 2022a; Butoi et al., 2023). Various approaches have been proposed, including image-level data augmentation techniques like style diversification (Liu et al., 2020b; Su et al., 2021a; Zhang et al., 2020; Su et al., 2022b), and feature space manipulation methods based on adversarial learning or statistical randomization (Chen et al., 2022; Zhou et al., 2022b; Chen et al., 2023; Ouyang et al., 2022a). Moreover, disentanglement-based strategies have been explored to promote cross-domain feature invariance (Hu et al., 2023; Gu et al., 2023). Recently, spectral decomposition has been shown to effectively capture style and content information, which are closely associated with amplitude and phase spectra (Liu et al., 2021; Xu et al., 2021; Wang et al., 2023b), in the context of DG. This observation motivates our investigation into the efficient transfer of generalization features from VFMs to lightweight specialized models within the frequency domain.

**Single-source Domain Generalization (SDG).** SDG methods (Fan et al., 2021; Liu et al., 2022b) focus on extracting robust and invariant features solely from source data. Traditional approaches include image transformations (Zhang et al., 2020), adversarial learning (Jing et al., 2023; Su et al., 2021b), model-driven augmentation (Yue et al., 2019), and feature-level augmentation (Zhou et al., 2022a). To mitigate overfitting risks caused by domain shifts, a variety of data augmentation strategies have been proposed (Xu et al., 2020; Huang et al., 2020; Zhou et al., 2021). Adversarial techniques (Zhong et al., 2022; Chen et al., 2020; Qiao et al., 2020) leverage domain synthesizers to generate interpolated domains while maintaining semantic consistency through mutual information regularization.

**Vision Foundation Model (VFM).** Foundation models such as SAM (Kirillov et al., 2023b), DINOV2 (Oquab et al., 2023), CLIP (Radford et al., 2021), GIT (Wang et al., 2022), and Coca (Yu et al., 2022) represent notable breakthroughs. These large-scale foundation models, pre-trained on diverse datasets, have attracted considerable attention for their flexibility in adapting to a wide range of tasks (Devlin et al., 2018; Touvron et al., 2023; Brown et al., 2020). Their re-

markable generalization abilities have inspired extensive research into their potential for specialized applications, demonstrating their transformative influence across various domains (Ji et al., 2023; Ma & Wang, 2023; Wang et al., 2023a; Osco et al., 2023). This inspires us to consider how to fully leverage these properties during the training phase.

## 3. Method

The proposed TinyMIG framework leverages VFMs such as SAM (Kirillov et al., 2023a), Dinov2 (Oquab et al., 2023), and SAMed (Ma et al., 2024) during the training phase to guide and enhance the single-domain medical image segmentation capabilities of lightweight models. The overall framework comprises two key components: Global Distribution Consistency Learning and Localized Representation Alignment. All strategies are implemented during the training phase, ensuring no additional computational overhead during the inference phase.

### 3.1. Global Distribution Consistency Learning

Pre-training on large-scale datasets with diverse styles endows VFMs with superior generalization capabilities in global feature distribution, enabling them to adapt to a wide variety of patterns. To leverage this advantage, we design the Universal Feature Distribution Modulation (UFDM), which adjusts the feature distribution of the lightweight model using the prior distribution of the foundation model. Additionally, we propose the Cross-Distribution Consistency Loss ($\mathcal{L}_{CDC}$) to optimize the consistency of the lightweight model's feature distribution during the inference.

**Universal Feature Distribution Modulation (UFDM).** We observe that the statistical properties of feature maps, such as mean and variance within a deep network, serve as strong indicators for features with varying styles and content. Inspired by the instance normalization strategy in AdaIN (Huang & Belongie, 2017), we propose the UFDM module to modulate the feature distributions within the specialized model, as demonstrated in Fig. 3 (a). Specifically, for a given intermediate feature $\mathcal{X}_i^{\mathcal{R}}$ and $\mathcal{X}_i^{\mathcal{S}} \in \mathbb{R}^{H \times W \times C}$ from the $i$-th block of the specialized and foundation models, $H$, $W$, and $C$ denote the height, width, and number of channels, respectively, and $\mathcal{X}_i^{\mathcal{R}}$, which is obtained by reprogramming the original foundation model features $\mathcal{X}_i^{\mathcal{F}}$ through the FRM module, as detailed in Section 3.4.

$$\text{AdaIN}(f, \gamma_m, \beta_m) = \gamma_m(f - \mu(f))/\sigma(f) + \beta_m, \quad (1)$$

where $\mu(\cdot), \sigma(\cdot) \in \mathbb{R}^{B \times C}$ correspond to the spatially computed mean and standard deviation for each channel across the features $f$, $\gamma_m$ and $\beta_m$ can determine the direction of style transfer. We begin by sampling $\mathcal{Q} \in \mathbb{R}^{B \times C}$ from a Beta distribution, $\mathcal{Q} \sim \text{Beta}(\alpha, \alpha)$, where $\alpha$ is empirically set to 0.1 (Zhou et al., 2021). The sampled $\mathcal{Q}$ is then used as

the probability parameter to define a Bernoulli distribution, from which we draw $\lambda \in \mathbb{R}^{B \times C}$, *i.e.*, $\lambda \sim \text{Bern}(\mathcal{Q})$. To ensure a better imitation of global features, we randomly sample the augmented statistics $\sigma_s, \mu_s \in \mathbb{R}^{B \times C}$ from a uniform distribution and mix them with the feature distribution of the foundation model. This distribution spans most feature statistics: $\sigma_s \sim U(0, 1), \mu_s \sim U(0, 1)$. Then, the mixed statistics are computed as follows:

$$\gamma_{\text{mix}} = \lambda \sigma_s + (1 - \lambda)\sigma(\mathcal{X}_i^{\mathcal{R}}), \beta_{\text{mix}} = \lambda \mu_s + (1 - \lambda)\mu(\mathcal{X}_i^{\mathcal{R}}). \quad (2)$$

Finally, the mixed feature statistics is applied to mimic the global distribution of the foundation features $\mathcal{X}_i^{\mathcal{R}}$:

$$UFDM(\mathcal{X}_i^S) = \gamma_{\text{mix}} \frac{\mathcal{X}_i^S - \mu(\mathcal{X}_i^S)}{\sigma(\mathcal{X}_i^S)} + \beta_{\text{mix}}. \quad (3)$$

**Cross-Distribution Consistency Loss (CDCL).** We introduce the $\mathcal{L}_{CDC}$ loss, which enhances domain-invariant feature extraction by aligning the soft predictions of the original and UFDM-modulated samples. Specifically, we adopt the logit pairing method, where CDCL enforces consistency by aligning segmentation logit outputs before and after distribution modulation during training. Formally, we minimize the bidirectional KL divergence between the probabilistic distributions of the semantic predictions, $\mathcal{Y}_i^{ufdm}$ and $\mathcal{Y}_i^{ori}$, ensuring that the specialized model maintains distributional consistency under different styles:

$$\begin{aligned} \mathcal{L}_{CDC} = &\frac{1}{N} \sum_{i=0}^{N-1} \left( \mathcal{KL} \left( \mathcal{Y}_i^{ori} \| \mathcal{Y}_i^{ufdm} \right) \right) \\ &+ \frac{1}{N} \sum_{i=0}^{N-1} \left( \mathcal{KL} \left( \mathcal{Y}_i^{ufdm} \| \mathcal{Y}_i^{ori} \right) \right). \end{aligned} \quad (4)$$

The $\mathcal{L}_{CDC}$ is designed to enhance the specialized model's ability to capture high-quality feature representations, thereby improving its performance in domain-specific tasks without relying on the foundation model during inference.

### 3.2. Localized Representation Alignment (LRA)

As shown in Fig. 3 (b), we design the Frequency-based Discriminative Content Enhancement and the Localized Representation Alignment Loss. These components work together to refine local feature representations, ensuring improved robustness in unseen domains.

**Frequency-based Discriminative Content Enhancement (FDCE).** To enhance domain-invariant and discriminative content in the frequency domain, we employ the Fourier Transform (FFT) to decompose feature representations from both the foundation and specialized models, followed by learnable filtering operations on amplitude and phase components. Specifically, given a pair of intermediate feature representations $\mathcal{X}_i^{\mathcal{S}}$ and reprogrammed features $\mathcal{X}_i^{\mathcal{R}} \in$

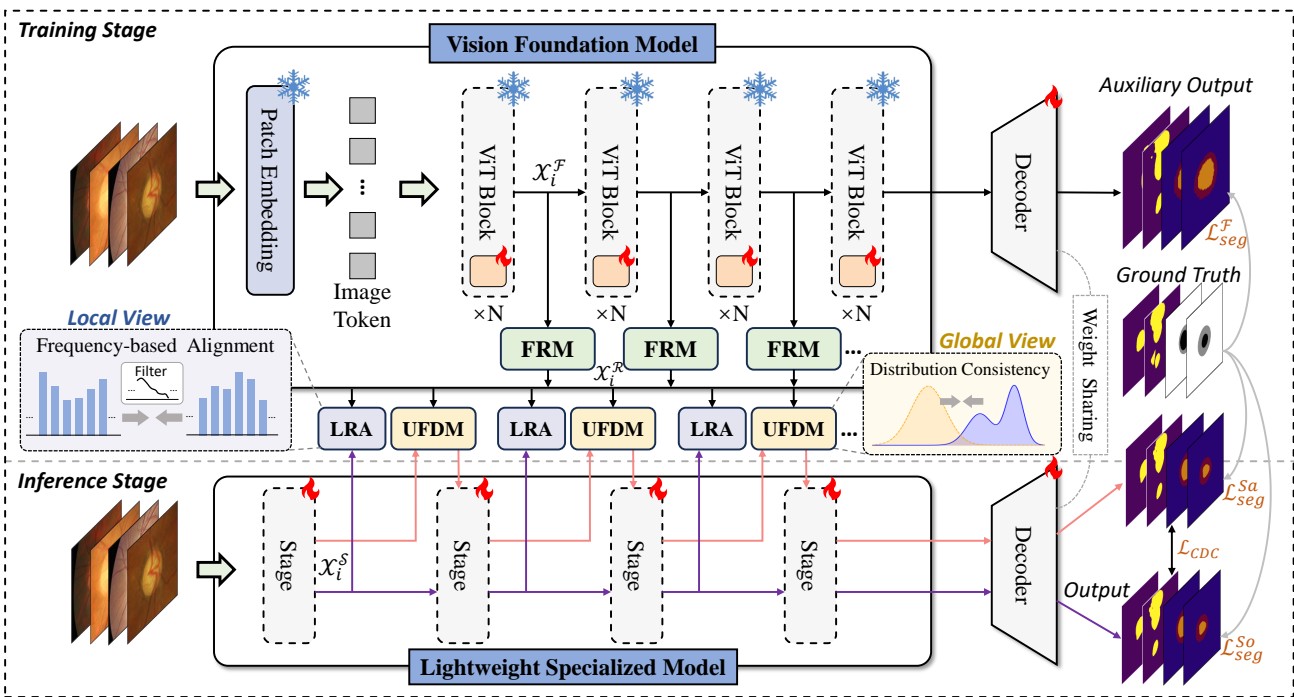

*Figure 2.* Illustration of the Proposed TinyMIG Framework. UFDM is integrated into certain layers of the specialized model to generate style samples. LRA enhances content features of both models in the frequency domain. The model is further optimized with $\mathcal{L}_{CDC}$ and $\mathcal{L}_{LRA}$, promoting the global and local transfer of generalized features to the specialized model. Segmentation loss constrains the outputs of both original $\mathcal{L}_{seg}^{\mathcal{S}_o}$ and UFDM-stylized features $\mathcal{L}_{seg}^{\mathcal{S}_a}$, while the foundation model $\mathcal{L}_{seg}^{\mathcal{F}}$ is incorporated via co-training. During inference, only the low-cost specialized model is used, ensuring a balance between generalization performance and efficiency.

$\mathbb{R}^{H \times W \times C}$ extracted from the $i$-th block of the specialized and foundation models via the FRM module, we first apply a 2D FFT (Chi et al., 2020) to obtain the corresponding frequency representations $\mathcal{X}_i^{\mathcal{S}f}$ and $\mathcal{X}_i^{\mathcal{R}f}$. As shown in the following equations, these frequency representations are then decomposed into amplitude and phase components:

$$[\mathcal{X}_i^{\mathcal{R}\mathcal{A}}, \mathcal{X}_i^{\mathcal{S}\mathcal{A}}] = \mathcal{A}([\mathcal{X}_i^{\mathcal{R}f}, \mathcal{X}_i^{\mathcal{S}f}]), \quad (5)$$

$$[\mathcal{X}_i^{\mathcal{R}\mathcal{P}}, \mathcal{X}_i^{\mathcal{S}\mathcal{P}}] = \mathcal{P}([\mathcal{X}_i^{\mathcal{R}f}, \mathcal{X}_i^{\mathcal{S}f}]). \quad (6)$$

To selectively enhance the domain-invariant and discriminative frequency components, we introduce two learnable filters, $\mathcal{W}_{\mathcal{A}}$ and $\mathcal{W}_{\mathcal{P}}$ (denoted as FDCE-A and FDCE-P), which are applied separately to the amplitude and phase spectra. The filtering operations are defined as follows:

$$[\hat{\mathcal{X}}_i^{\mathcal{R}\mathcal{P}}, \hat{\mathcal{X}}_i^{\mathcal{S}\mathcal{P}}] = \text{Sigmoid}(\mathcal{W}_{\mathcal{P}}) \otimes [\mathcal{X}_i^{\mathcal{R}\mathcal{P}}, \mathcal{X}_i^{\mathcal{S}\mathcal{P}}], \quad (7)$$

$$[\hat{\mathcal{X}}_i^{\mathcal{R}\mathcal{A}}, \hat{\mathcal{X}}_i^{\mathcal{S}\mathcal{A}}] = \text{Sigmoid}(\mathcal{W}_{\mathcal{A}}) \otimes [\mathcal{X}_i^{\mathcal{R}\mathcal{A}}, \mathcal{X}_i^{\mathcal{S}\mathcal{A}}], \quad (8)$$

where $\otimes$ denotes element-wise multiplication, and the enhanced phase and amplitude representations are denoted as $\hat{\mathcal{X}}_i^{\mathcal{R}\mathcal{P}}$ and $\hat{\mathcal{X}}_i^{\mathcal{R}\mathcal{A}}$, respectively. The learnable filters are initialized with all elements set to one, ensuring that $\text{Sigmoid}(\mathcal{W}_*) \in [0, 1]$. Finally, the enhanced amplitude and phase spectra are reconstructed into feature maps using the

inverse Fourier Transform ($F^{-1}$), allowing the specialized model to leverage the enriched frequency-domain representations:

$$\{\hat{\mathcal{X}}_i^{\mathcal{R}}, \hat{\mathcal{X}}_i^{\mathcal{S}}\} = F^{-1}\{[\hat{\mathcal{X}}_i^{\mathcal{R}\mathcal{P}}, \hat{\mathcal{X}}_i^{\mathcal{R}\mathcal{A}}], [\hat{\mathcal{X}}_i^{\mathcal{S}\mathcal{P}}, \hat{\mathcal{X}}_i^{\mathcal{S}\mathcal{A}}]\}. \quad (9)$$

After applying spectral filters for content enhancement at the spatial locations, we further propose a phase spectrum-based Adaptive Channel Refiner to enhance content along the channel dimension. The key idea is that amplitude reflects style, whereas phase represents content. Thus, the phase component is assumed to be consistent between the foundation and specialized features $\mathcal{X}_i^{\mathcal{R}}$ and $\mathcal{X}_i^{\mathcal{S}}$ due to its stable representation. Subsequently, we obtain the refined channel weight $\mathcal{C}_{\mathcal{P}}^{\mathcal{R}}$ and $\mathcal{C}_{\mathcal{P}}^{\mathcal{S}}$ through applying SE blocks (Hu et al., 2018) to the phase element of the content enhanced features $\hat{\mathcal{X}}_i^{\mathcal{S}}$ and $\hat{\mathcal{X}}_i^{\mathcal{R}}$, as shown below:

$$\mathcal{C}_{\mathcal{P}}^{\mathcal{R}} = SE(\hat{\mathcal{X}}_i^{\mathcal{R}\mathcal{P}}), \mathcal{C}_{\mathcal{P}}^{\mathcal{S}} = SE(\hat{\mathcal{X}}_i^{\mathcal{S}\mathcal{P}}). \quad (10)$$

Then we product the channel attention with $\mathcal{X}_i^{\mathcal{S}}$ and $\mathcal{X}_i^{\mathcal{R}}$ to obtain the final refined feature map $\tilde{\mathcal{X}}_i^{\mathcal{S}}$ and $\tilde{\mathcal{X}}_i^{R}$:

$$\tilde{\mathcal{X}}_i^{\mathcal{R}} = \phi_{ex}(\mathcal{C}_{\mathcal{P}}^{\mathcal{R}}) \odot \hat{\mathcal{X}}_i^{\mathcal{R}}, \tilde{\mathcal{X}}_i^{\mathcal{S}} = \phi_{ex}(\mathcal{C}_{\mathcal{P}}^{S}) \odot \hat{\mathcal{X}}_i^{\mathcal{S}}, \quad (11)$$

where $\odot$ denotes the Hadamard product, and $\phi_{ex}(\cdot)$ adapts the weights to align with the spatial dimensions of the feature map, i.e., $\phi_{ex} : \mathbb{R}^{c \times 1 \times 1} \to \mathbb{R}^{c \times h \times w}$.

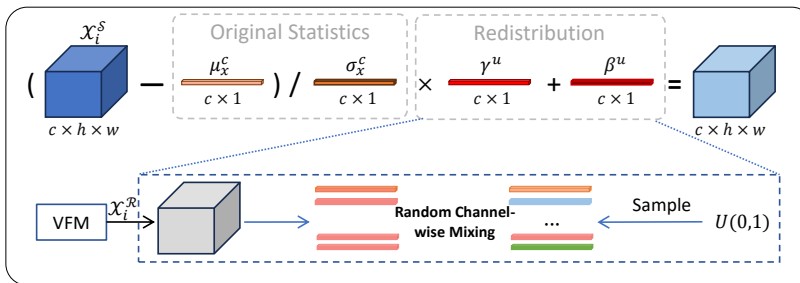
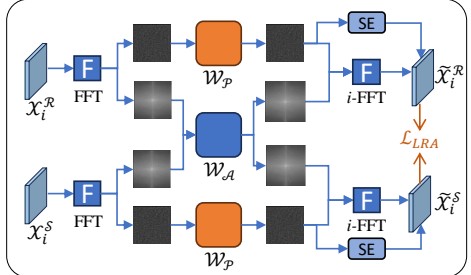

(a) Universal Feature Distribution Modulation (b) The proposed FDCE module

*Figure 3.* (a) The proposed Universal Feature Distribution Modulation (UFDM) module, by extracting feature distributions $\mathcal{X}_i^{\mathcal{R}}$ and $\mathcal{X}_i^{\mathcal{S}}$ from the foundation model and specialized model, UFDM generates the redistributed samples. (b) Frequency-based Discriminative Content Enhancement (FDCE) module utilizes Fourier transforms combined with learnable frequency-domain filters $\mathcal{W}_{\mathcal{A}}$ and $\mathcal{W}_{\mathcal{P}}$ to enhance the discriminative content features.

**Localized Representation Alignment Loss**. We implement the FDCE strategy across multiple feature scales to enhance feature representations at different hierarchical levels. Specifically, we select a set of intermediate feature maps from both the foundation model and the specialized model:

$$\tilde{\mathcal{X}}^{\mathcal{R}} = \{\tilde{\mathcal{X}}_i^{\mathcal{R}} \mid i \in \mathcal{I}_R\}, \quad \tilde{\mathcal{X}}^{\mathcal{S}} = \{\tilde{\mathcal{X}}_j^{\mathcal{S}} \mid j \in \mathcal{I}_S\}, \quad (12)$$

where $\mathcal{I}_R$ and $\mathcal{I}_S$ represent the selected sets of feature layers from the foundation and specialized models, respectively. To enforce consistency between the feature representations, we introduce an L2-norm loss that minimizes the discrepancy between corresponding feature maps:

$$\mathcal{L}_{LRA} = \frac{1}{N} \sum_{c=1}^{C} \sum_{h=1}^{H} \sum_{w=1}^{W} \left( \tilde{\mathcal{X}}_i^{\mathcal{R}}(c,h,w) - \tilde{\mathcal{X}}_j^{\mathcal{S}}(c,h,w) \right)^2 . \tag{13}$$

### 3.3. PEFT for Foundation Model

To enhance the adaptability of vision foundation models (VFMs) to the medical domain and improve feature alignment with the specialized model, we adopt a two-step approach. First, we employ the Adapter technique, as used in DAPSAM (Wei et al., 2024b), which enables Parameter-Efficient Fine-Tuning at each layer of the foundation model. Second, we propose the Feature Reprogramming Module (FRM) to further refine the extracted features, making them more suitable for downstream tasks and specialized models. At its core, FRM utilizes residual blocks to perform feature dimension transformation and alignment, ensuring effective adaptation of foundation model representations.

**Training Objective.** In this work, two distinct loss functions are employed: auxiliary segmentation loss $\mathcal{L}_{seg}^{\mathcal{F}}$ for the VFM and the segmentation loss $\mathcal{L}_{seg}^{\mathcal{S}}$ for the specilized model. $\mathcal{L}_{seg}$ comprises a combination of Dice loss and Cross-Entropy loss. Thus, our overall optimization objective can be expressed by the following equation:

$$\mathcal{L} = \lambda_1 \mathcal{L}_{seg}^{\mathcal{F}} + \lambda_2 \mathcal{L}_{seg}^{Sa} + \lambda_3 \mathcal{L}_{seg}^{So} + \lambda_4 \mathcal{L}_{LRA} + \lambda_5 \mathcal{L}_{CDC}, \tag{14}$$

where $[\lambda_1,...,\lambda_5]$ are hyperparameters. A detailed discussion of loss term hyperparameters is in the Appendix.

## 4. Experiment

To validate the effectiveness of our TinyMIG framework, we conduct extensive experiments on two medical image DG benchmark tasks. These include two medical segmentation tasks: the 2D joint optic disc (OD) and cup (OC) segmentation task (Almazroa et al., 2018a; Chen et al., 2023), as well as a 3D medical image segmentation task: the prostate MRI segmentation task (Chen et al., 2023). The Dice score metric (DSC) and Average Surface Distance (ASD) are utilized for evaluation on fundus and prostate task. We closely follow the implementation of (Hu et al., 2023) and use a U-shaped segmentation network with a modified ResNet-34 encoder as the specialized model and backbone for all competing approaches, ensuring a fair comparison. Further details on the dataset and model architecture are provided in the appendix.

### 4.1. Experimental Results

**Experiment Setting.** In our experiments, we adopt the leave-one-domain-out strategy commonly used in DG research. The model is trained on the single source domain and tested on the remaining $K-1$ unseen target domains. We compare with several recent state-of-the-art domain generalization approaches, including: feature-space domain randomization Methods: MixStyle (Zhou et al., 2021), TriD (Chen et al., 2023), and DSU (Li et al., 2022); Adversarial noise synthesis method: MaxStyle (Chen et al., 2022); Image augmentation methods for SDG in Medical Imaging: SLAug (Su et al., 2022b) and VPTTA (Chen et al., 2024). Advanced DG methods based on vision foundation models (*e.g.*, fine-tuned VFs): Rein (Wei et al., 2024a) and DAPSAM (Wei et al., 2024b).

**Comparison results:** Tab. 1 and Tab. 2 present the quantitative comparison results of our TinyMIG and other DG

*Table 1.* Performance Comparison of our TinyMIG with SOTA methods on Fundus segmentation task.

| Methods | Optical Disc / Cup Segmentation (DSC ↑) | | | | | Avg. ↑ | Optical Disc / Cup Segmentation (DSC ↑) | | | | | Avg. ↑ |
|---|---|---|---|---|---|---|---|---|---|---|---|---|
| | $\mathcal{D}_1$ | $\mathcal{D}_2$ | $\mathcal{D}_3$ | $\mathcal{D}_4$ | $\mathcal{D}_5$ | | $\mathcal{D}_1$ | $\mathcal{D}_2$ | $\mathcal{D}_3$ | $\mathcal{D}_4$ | $\mathcal{D}_5$ | |
| | Optical Disc DSC ↑ | | | | | | Optical Cup DSC ↑ | | | | | |
| ERM (Ronneberger et al., 2015) | 74.89 | 81.72 | 78.12 | 74.79 | 74.36 | 76.78 | 59.21 | 63.27 | 71.96 | 57.23 | 58.88 | 62.11 |
| MixStyle (ICLR2021) (Zhou et al., 2021) | 75.67 | 83.35 | 82.86 | 68.86 | 79.54 | 78.06 | 60.84 | 62.60 | 73.77 | 61.44 | 66.79 | 66.73 |
| CSDG (TMI 2022) (Ouyang et al., 2022a) | 78.40 | 82.02 | 81.46 | 75.51 | 81.09 | 79.70 | 65.11 | 70.79 | 76.19 | 65.26 | 65.28 | 68.53 |
| MaxStyle (MICCAI2022) (Chen et al., 2022) | 77.40 | 80.95 | 79.59 | 76.69 | 81.95 | 79.32 | 65.44 | 67.62 | 74.52 | 66.05 | 64.84 | 67.10 |
| EFDM (CVPR 2022) (Zhang et al., 2022) | 78.83 | 84.83 | 82.25 | 82.13 | 81.45 | 81.90 | 62.75 | 65.94 | 72.20 | 61.62 | 63.02 | 64.10 |
| DSU (ICLR2022) (Li et al., 2022) | 76.88 | 82.17 | 81.12 | 82.36 | 83.09 | 81.12 | 61.26 | 70.16 | 74.10 | 63.19 | 59.65 | 65.67 |
| SLAug (AAAI2023) (Su et al., 2022b) | 79.83 | 83.42 | 83.18 | 81.17 | 83.57 | 82.23 | 64.53 | 71.30 | 75.94 | 64.52 | 67.12 | 68.28 |
| TriD (MICCAI 2023) (Chen et al., 2023) | 78.35 | 82.19 | 83.62 | 80.18 | 81.65 | 81.12 | 66.67 | 70.85 | 74.13 | 67.53 | 66.96 | 69.23 |
| MAD (CVPR2023) (Qu et al., 2023) | 80.63 | 82.10 | 80.37 | 82.08 | 80.31 | 81.10 | 67.14 | 66.57 | 72.40 | 68.38 | 69.37 | 68.77 |
| VPTTA (CVPR2024) (Chen et al., 2024) | 79.41 | 84.86 | 80.01 | 62.01 | 80.85 | 77.43 | 68.41 | 73.86 | 69.01 | 51.01 | 69.85 | 66.43 |
| DAPSAM (MICCAI2024) (Wei et al., 2024b) | 85.36 | 89.01 | 85.45 | 87.29 | 90.38 | 88.27 | 76.38 | 76.80 | 77.14 | 72.62 | 72.01 | 74.99 |
| SAMed (Nat. Commun2024) (Ma et al., 2024) | 80.63 | 82.10 | 80.37 | 82.08 | 80.31 | 81.10 | 67.14 | 66.57 | 72.40 | 68.38 | 69.37 | 68.77 |
| ***Tiny-MIG(Ours)*** | **90.83** | **89.38** | **90.01** | **88.86** | **91.36** | **90.09** | **82.60** | **79.14** | **81.35** | **76.83** | **79.26** | **79.84** |
| | +5.47 | +0.37 | +4.56 | +1.57 | +0.98 | +1.82 | +6.22 | +2.34 | +4.21 | +4.21 | +7.25 | +4.85 |

*Table 2.* Performance Comparison of our TinyMIG with SOTA methods on Prostate segmentation task.

| Task | Prostate Segmentation (DSC ↑) | | | | | | Avg. ↑ | Prostate Segmentation (ASD ↓) | | | | | | Avg. ↓ |
|---|---|---|---|---|---|---|---|---|---|---|---|---|---|---|
| Single Seen Site | $\mathcal{D}_1$ | $\mathcal{D}_2$ | $\mathcal{D}_3$ | $\mathcal{D}_4$ | $\mathcal{D}_5$ | $\mathcal{D}_6$ | | $\mathcal{D}_1$ | $\mathcal{D}_2$ | $\mathcal{D}_3$ | $\mathcal{D}_4$ | $\mathcal{D}_5$ | $\mathcal{D}_6$ | |
| | DSC ↑ | | | | | | | ASD ↓ | | | | | | |
| ERM (Ronneberger et al., 2015) | 71.81 | 65.56 | 43.98 | 71.97 | 48.39 | 37.82 | 56.59 | 7.54 | 8.87 | 13.30 | 11.97 | 9.98 | 7.65 | 9.89 |
| MixStyle (Zhou et al., 2021) | 73.24 | 58.06 | 44.75 | 66.78 | 49.81 | 49.73 | 57.06 | 4.98 | 5.77 | 6.30 | 5.21 | 5.98 | 6.26 | 5.75 |
| CSDG (Ouyang et al., 2022b) | 82.14 | 67.21 | 59.11 | 73.16 | 67.38 | 73.23 | 70.37 | 3.51 | 4.08 | 4.56 | 3.58 | 4.46 | 4.17 | 4.06 |
| CCSDG (Guo et al., 2023) | 80.62 | 69.52 | 65.18 | 67.89 | 58.99 | 63.27 | 67.58 | 3.76 | 4.12 | 4.68 | 3.61 | 4.42 | 4.87 | 4.24 |
| MaxStyle (Chen et al., 2022) | 81.25 | 70.27 | 62.09 | 58.18 | 70.04 | 67.77 | 68.27 | 3.40 | 3.80 | 4.32 | 3.23 | 3.67 | 4.12 | 3.77 |
| EFDM (Zhang et al., 2022) | 80.87 | 69.78 | 63.16 | 65.39 | 69.84 | 67.15 | 69.37 | 3.45 | 3.82 | 4.35 | 3.37 | 3.89 | 4.03 | 3.82 |
| SLAug (Su et al., 2022b) | 81.20 | 69.32 | 60.92 | 73.72 | 67.15 | 71.93 | 70.71 | 3.31 | 3.74 | 4.23 | 3.22 | 3.79 | 3.91 | 3.70 |
| TriD (Chen et al., 2023) | 81.50 | 70.28 | 62.89 | 74.52 | 72.12 | 69.11 | 71.74 | 3.28 | 3.69 | 4.15 | 3.14 | 3.67 | 3.81 | 3.62 |
| MAD (Qu et al., 2023) | 80.87 | 69.78 | 63.16 | 65.39 | 69.84 | 67.15 | 69.37 | 3.49 | 3.81 | 4.33 | 3.36 | 3.87 | 4.01 | 3.82 |
| VPTTA (Chen et al., 2024) | 82.32 | 71.12 | 66.89 | 76.31 | 76.98 | 73.87 | 74.58 | 3.92 | 3.43 | 4.65 | 3.08 | 3.59 | 3.85 | 3.75 |
| DAPSAM (Wei et al., 2024b) | **86.34** | 81.05 | 70.81 | 85.28 | 82.91 | 81.48 | 81.31 | 3.78 | 3.49 | 3.98 | 2.79 | 3.12 | 3.16 | 3.39 |
| SAMed (Ma et al., 2024) | 80.42 | 81.44 | 66.75 | 82.09 | 80.19 | 80.17 | 78.51 | 4.78 | 4.49 | 5.54 | 4.27 | 4.34 | 4.61 | 4.67 |
| ***TinyMIG (Ours)*** | 85.82 | **84.52** | **76.45** | **86.61** | **83.70** | **81.53** | **83.09** | **3.81** | **3.30** | **3.52** | **1.62** | **3.09** | **3.02** | **3.06** |
| | -0.52 | +3.47 | +5.64 | +1.33 | +0.79 | +0.05 | +1.79 | -0.03 | +0.19 | +0.46 | +1.17 | +0.02 | +0.14 | +0.33 |

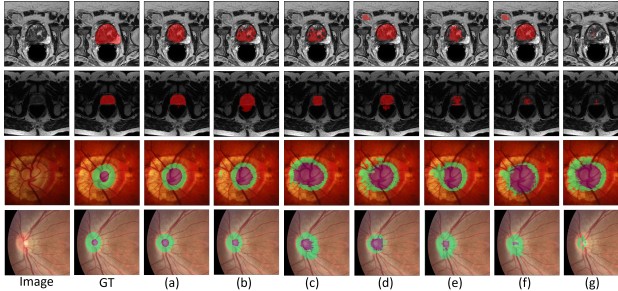

*Figure 4.* Comparisons across different SDG Methods: prostate imaging (The first two rows), fundus imaging (The last two rows) with ground truth (GT) and predictions (a-g). The subfigures (a) to (g) correspond to: (a) ***TinyMIG***, (b) DAPSAM (Wei et al., 2024b), (c) TriD (Chen et al., 2023), (d) VPTTA (Chen et al., 2024), (e) MAD (Qu et al., 2023), (f) SLAug (Su et al., 2022b), (g) baseline.

methods on the fundus segmentation task and the prostate MRI segmentation task, respectively. As shown in Tab. 1, our method achieves the best segmentation performance on the fundus segmentation task, with an average DSC score of 84.97% for OD/OC segmentation, surpassing the recent state-of-the-art DAPSAM by 3.92%, demonstrating the effectiveness of our approach. Tab. 2 compares prostate MRI segmentation performance across methodologies. Notably, foundation model-based fine-tuning approaches (*e.g.*, Rein,

DAPSAM) surpass specialized domain generalization methods, demonstrating foundation models' superior capacity to address cross-domain shifts through enhanced feature representations. Our approach delivers consistent gains across metrics, surpassing DAPSAM (Wei et al., 2024b) (DSC: +1.78%, ASD: -0.59) and achieving remarkable 5.64% DSC improvement under Domain 3's challenging settings. Crucially, while maintaining computational efficiency (22.6 GFLOPs), our framework outperforms foundation model fine-tuning by eliminating the substantial inference overhead associated with foundation models, while achieving an optimal accuracy-efficiency tradeoff (Fig. 1). Fig. 4 presents segmentation results for two cases from unseen domains across different tasks. Our method accurately delineates organ structures and boundaries in images from unknown distributions, whereas other methods often confuse organ boundaries with surrounding tissues, leading to segmentation errors.

**Various VFMs in TinyMIG.** We select three VFMs: Med-SAM (base) (Ma et al., 2024), SAM (base) (Kirillov et al., 2023a), and DINOv2 (vit_base) (Oquab et al., 2023). Tab. 3's last three rows show that all VFMs significantly boost segmentation accuracy on medical imaging tasks. MedSAM outperforms others, likely due to its medical image-specific

*Table 3.* Comparison of different TinyMIG variants and fine-tuning-based methods in terms of segmentation performance (Dice ↑), parameter count (M), throughout (fps), and computational cost (G-FLOPs) during the testing phase. All experiments are conducted on an NVIDIA RTX 4090 GPU.

| Methods | prostate | fundus | Params. | throughout | FLOPs |
|---|---|---|---|---|---|
| DAPSAM | 81.31 | 81.05 | 98.45M | 43.91 | 58.61 |
| Rein+SAM | 79.85 | 79.36 | 95.78M | 44.10 | 54.29 |
| Rein+Dinov2 | 80.07 | 79.54 | 86.96M | 45.72 | 50.72 |
| MedSAM | 78.51 | 75.92 | 93.74M | 44.92 | 52.39 |
| **TinyMIG+SAMed** | **83.09** | **84.68** | **22.52M** | **147.73** | **22.64** |
| **TinyMIG+SAM** | **82.69** | **83.57** | **22.52M** | **147.73** | **22.64** |
| **TinyMIG+Dinov2** | **82.53** | **83.05** | **22.52M** | **147.73** | **22.64** |

*Table 4.* Ablation experiments on each component in TinyMIG for the fundus and prostate tasks.

| UFDM | $\mathcal{L}_{CDC}$ | FDCE | $\mathcal{L}_{LRA}$ | Co-training | Fundus | Prostate |
|---|---|---|---|---|---|---|
| ✗ | ✗ | ✗ | ✗ | ✗ | 67.53 | 48.75 |
| ✓ | ✗ | ✗ | ✗ | ✓ | 81.04 | 76.25 |
| ✓ | ✓ | ✗ | ✗ | ✓ | 82.52 | 80.73 |
| ✗ | ✗ | ✓ | ✓ | ✓ | 83.41 | 78.26 |
| ✗ | ✗ | ✗ | ✓ | ✓ | 80.16 | 70.32 |
| ✓ | ✓ | ✓ | ✓ | ✗ | 83.78 | 81.76 |
| ✓ | ✓ | ✓ | ✓ | ✓ | **84.97** | **83.09** |

pretraining. While SAM and DINOv2 are trained on natural images, they still deliver strong segmentation results, demonstrating TinyMIG's robust generalization and feature transfer capabilities. Compared to DAPSAM, TinyMIG not only achieves superior segmentation performance on unseen domain samples but also offers significantly better inference efficiency: fewer model parameters (22.52 *v.s.* 98.45 M), faster inference speed (147.73 *v.s.* 43.92 fps), and reduced computational cost (22.64 *v.s.* 58.61 GFLOPs). These advantages validate the adaptability of TinyMIG method to different VFMs, demonstrating its generalizability.

## 4.2. Ablation Studies of TinyMIG

We conduct ablation studies to evaluate the contribution of each component in our TinyMIG as summarized in Tab. 4.

**Effectiveness of UFDM**. The UFDM module is the core of Global Distribution Consistency Learning. When using the UFDM module only, we directly supervise the stylized samples with the cross entropy loss. As shown in the second row of Tab. 4, UFDM significantly improves model performance even without the proposed knowledge-driven consistency loss. This highlights the importance of diversifying training samples. Meanwhile, we compare UFDM with other style-randomization-based DG methods, such as MixStyle (Zhou et al., 2021), DSU (Li et al., 2022) and Maxstyle (Chen et al., 2022), as illustrated in Tab. 5, UFDM achieves the best segmentation performance, further demonstrating the effectiveness of redistrbuting style features from vision foundation models.

*Table 5.* Ablation study of different style variation methods in UFDM on prostate MRI task.

| | Mixstyle | CrossNorm | DSU | Maxstyle | **TinyMIG** |
|---|---|---|---|---|---|
| **Dice** ↑ | 81.34 | 81.78 | 81.47 | 81.86 | 83.09 |
| **ASD** ↓ | 3.52 | 3.48 | 3.50 | 3.34 | 3.06 |

**Effectiveness of CDCL**. CDCL serves as the consistency learning component in the Global Distribution Consistency Learning method, aiming to guide the model in learning style-invariant representations. As shown in the third row of Tab. 4, compared with only applying UFDM, CDCL significantly improves the segmentation performance by 4.3% in DSC. This demonstrates that the proposed logit pairing over KL divergence loss effectively enables the model to leverage the generalization priors of foundational vision models, showcasing its superiority in learning domain-invariant representations.

**Effectiveness of FDCE**. As shown in the fourth row of Tab. 4, the absence of FDCE for enhancing domain-invariant content features leads to a significant performance drop. We also visualize feature heatmaps, as shown in Fig. 6, to further evaluate the effectiveness of the LRA module. In general, LRA in TinyMIG enables a more comprehensive focus on the organ structures and even edge details during segmentation. This is attributed to FDCE's spectral modulation in the frequency domain, which enhances the model's attention to details, edges, and richer texture information.

## 4.3. Further Evaluation

**Comparison of different style variation methods with UFDM.** In Tab. 5, we compare UFDM with MixStyle (Zhou et al., 2021), CrossNorm (Tang et al., 2021), DSU (Li et al., 2022), Maxstyle (Chen et al., 2022) and Trid (Chen et al., 2023). MixStyle and CrossNorm, being simple linear combinations of known source domain samples, struggle to synthesize rare styles. When using MixStyle and CrossNorm, their segmentation performance is 81.34% and 81.78%, respectively. In contrast, our UFDM method effectively integrates diverse style features from the visual backbone model, significantly increasing the style diversity of generated samples, which in turn enhances the model's segmentation performance.

**Ablation Analysis of FDCE.** We conduct the experiments shown in Tab. 6 to investigate the impact of different modules within the FDCE. The results indicate that removing any module negatively affects the segmentation performance on unseen domains, with FDCE-A having the most significant impact. This suggests that the amplitude spectrum enhancement filter plays a crucial role in the FDCE module, likely because the amplitude spectrum is closely associated with style and texture information of extracted features. Ad-

*Table 6.* Ablation study of LRA modules on prostate MRI task.

| FDCE-P | FDCE-A | ACR | Dice ↑ | ASD ↓ |
|:---:|:---:|:---:|:---:|:---:|
| ✗ | ✗ | ✗ | 56.59 | 9.89 |
| ✓ | ✗ | ✓ | 78.43 | 6.39 |
| ✗ | ✓ | ✓ | 81.02 | 3.81 |
| ✓ | ✓ | ✗ | 79.05 | 5.30 |
| ✓ | ✓ | ✓ | 83.09 | 3.06 |

*Table 7.* Comparison of Different Distillation Methods on prostate MRI task.

| | ChaKD | AttnFD | MLKD | **Ours** |
|:---:|:---:|:---:|:---:|:---:|
| **Dice ↑** | 81.02 | 81.78 | 80.56 | 83.09 |
| **ASD ↓** | 3.82 | 3.51 | 4.03 | 3.06 |

ditionally, since the phase spectrum encodes more domain-invariant content features, the ACR module plays a vital role in enhancing the content information.

**Comparison of Different LRA Methods**. We compare LRA with other distillation methods at the feature and logit level, including the channel distillation ChaKD (Shu et al., 2021), the attention-based distillation AttnKD (Komodakis & Zagoruyko, 2017) and logit distillation method MLKD (Jin et al., 2023). As shown in Tab. 7, our domain-invariant feature alignment method achieves the best performance compared to other approaches. This stems from the ability of spectral dynamic filtering to facilitate domain-invariant feature transfer while effectively avoids interference from domain-specific features in LRA module.

**Comparison of different alignment methods in CDCL.** Tab. 8 presents the impact of various consistency losses, including KL-Divergence (KL-Div), mean squared error (MSE), and Jensen–Shannon Divergence (JS-Div). The results indicate that the choice of consistency loss has minimal influence on our method's overall performance, demonstrating its robustness to different loss types.

**Feature Discriminability Visualization.** Fig. 5 presents t-SNE visualizations (Van der Maaten & Hinton, 2008) comparing feature discriminability between TinyMIG and the DAPSAM method (Chen et al., 2023). In the first column, our method effectively clusters samples from different domains within the same category, with well-defined distribution contours outlined by *magenta* and *violet* dashed lines. In contrast, the DAPSAM method exhibits a noticeable gap between them. The second column shows that DAPSAM struggles to distinguish category features effectively, likely due to its limited adaptability to OOD conditions and large style variations. In comparison, TinyMIG demonstrates clear separation across domains and categories, highlighting its strong generalization capability to unseen domains.

**Feature Visualization of LRA module.** To investigate the impact of the LRA module on enhancing domain-invariant content features, we visualize the heatmaps of segmentation models with and without the LRA module. As shown in

*Table 8.* Performance comparisons of different consistence learning method on prostate task.

| | MSE | KL-div | JS-div | **Ours** |
|:---:|:---:|:---:|:---:|:---:|
| **Dice ↑** | 81.02 | 81.78 | 82.56 | 83.09 |
| **ASD ↓** | 3.82 | 3.51 | 3.21 | 3.06 |

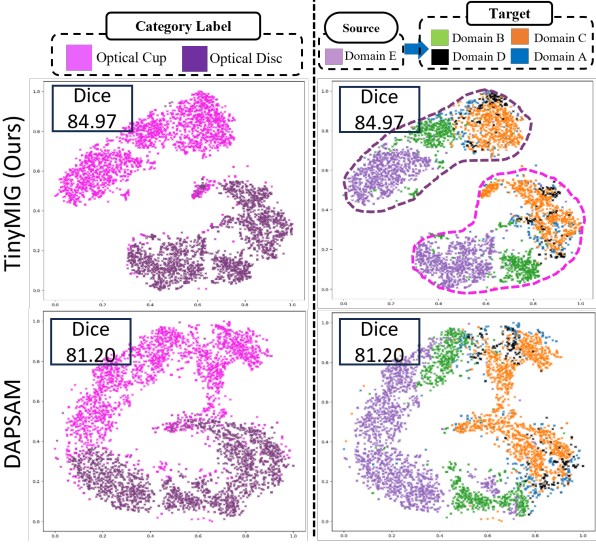

*Figure 5.* The t-SNE visualization uses colors to distinguish categories in the first column and domains in the second, with the *magenta* and *violet* dashed lines marking the distribution contours of two categories.

Fig. 6, the LRA module improves semantic focus, as seen in second column, which highlights more accurate regions. In contrast, the third column not only overlooks some semantic categories but also attends to irrelevant background areas. This demonstrates that the LRA module effectively enhances content features related to semantic categories.

## 5. Conclusion

We present **TinyMIG**, a novel framework that effectively transfers generalization from VFMs to compact architectures for medical imaging. TinyMIG introduces two synergistic components: Global Distribution Consistency Learning, which establishes hierarchical feature alignment across domains through multi-scale distribution matching while preserving essential anatomical structures, and Localized Representation Alignment, which enhances pathological detail capture through adaptive frequency filtering and discriminative feature refinement. This dual mechanism enables comprehensive learning of both global tissue organization and localized lesion patterns with remarkable computational efficiency. Extensive experiments with different VFMs validate that TinyMIG achieves superior performance while requiring significantly fewer computational resources than existing methods, establishing a new paradigm for developing efficient specialized models in medical image analysis.

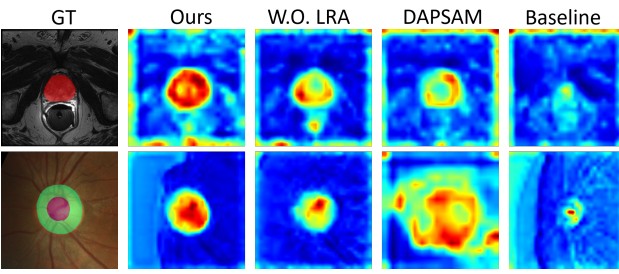

*Figure 6.* Heatmap visualization of the features from the encoder of the specilized model enhanced by LRA on two medical samples.

## Acknowledgments

This work was supported in part by the National Key Research and Development Program of China under Grant No.2021ZD0140407, the Beijing Natural Science Foundation L222152, the National Natural Science Foundation of China under Grant No.U21A20523 and National Natural Science Foundation of China under Grant No.62406347.

## Impact Statement

This paper presents work whose goal is to advance the field of Machine Learning. There are many potential societal consequences of our work, none which we feel must be specifically highlighted here.

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

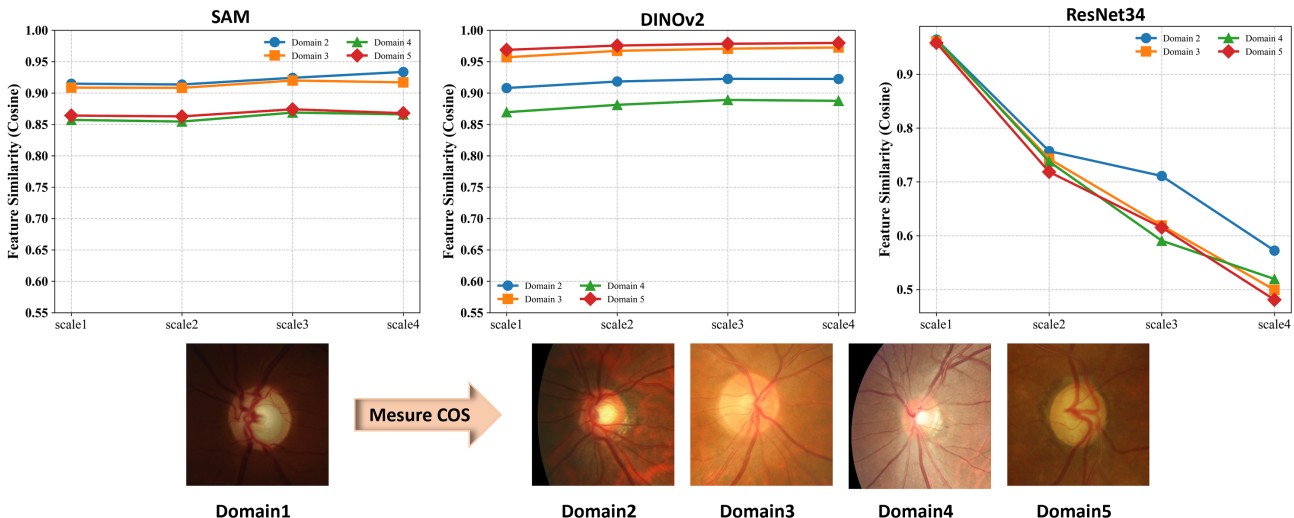

*Figure 7.* Feature Similarity Analysis of Backbone Models Across Different Architectures (SAM, DINOv2, and ResNet). We calculate the cosine similarity between features extracted at different layers of the backbone networks for images from various domains in the fundus dataset. Scale 1 refers to the first four ViT blocks of SAM (Kirillov et al., 2023a) and DINOv2 (Oquab et al., 2023).

The supplementary materials are organized as follows. In the section A, we analyze the motivation behind our proposed method. In the section B, we provide a detailed description of the dataset and the data augmentation techniques used. Section C presents extensive additional experiments to demonstrate the method's effectiveness.

## A. Motivation and Generalization of the Foundation Model.

In this section, we investigate whether the encoded features of vision foundation models trained on large-scale visual image datasets exhibit properties beneficial for generalization. We select two vision foundation models: SAM-b and Dinov2 (ViT-B), alongside a specialized model, ResUnet34, to evaluate their cosine similarity across different domain data on the fundus task. We examine the four-layer features from SAM and Dinov2 encoders: $[\mathcal{X}_3^{\mathcal{R}}, \mathcal{X}_6^{\mathcal{R}}, \mathcal{X}_9^{\mathcal{R}}, \mathcal{X}_{12}^{\mathcal{R}}]$, and the four-layer features from the ResUnet34 encoder: $[\mathcal{X}_1^{\mathcal{S}}, \mathcal{X}_2^{\mathcal{S}}, \mathcal{X}_3^{\mathcal{S}}, \mathcal{X}_4^{\mathcal{S}}]$. To assess the robustness of these features, data from domain$_1$ and domain$_{2-5}$ are input into the frozen SAM-b, Dinov2, and ResUnet34 encoders, producing the corresponding encoded features at each scale: $\mathcal{R}_i^{domain_k}$ and $\mathcal{S}_j^{domain_k}$, where i and j represent the different scale for VFMs (layer 3,6,9,12) and lightweight specialized models (layer 1,2,3,4), k depends different domains. Cosine similarities between the domain1 features and features from other domains are then computed for each scale. The results, presented in Fig. 7, show that the cosine similarities of features from SAM and Dinov2 remain highly stable across different domains, with strong similarity observed even in the deeper scales. In contrast, the lightweight specialized model, ResNet34, demonstrates a decreasing similarity as the network depth increases. From these findings, we conclude that the features from the VFMs' encoders are robust and distortion-invariant, which is crucial for developing transferable models with strong generalization capabilities.

## B. Datasets and Experimental Details

### B.1. Datasets Details

To validate the effectiveness and versatility of the proposed method, we conduct extensive experiments on three benchmark tasks: the joint Optic Disc (OD) and Cup (OC) segmentation task, the polyp segmentation task and the prostate segmentation task.

**The OD/OC segmentation task** comprises five public datasets collected from different medical centres, denoted as $\mathcal{D}_1$ (RIM-ONE-r3 (Orlando et al., 2020)), $\mathcal{D}_2$ (REFUGE(Almazroa et al., 2018b)), $\mathcal{D}_3$ (ORIGA (Zhang et al., 2010)), $\mathcal{D}_4$ (REFUGE-Validation/Test (Almazroa et al., 2018b)), and $\mathcal{D}_5$ (Drishti-GS (Sivaswamy et al., 2014)). There are 159, 400, 650, 800, and 101 images from these datasets. We cropped a region of interest (ROI) centering at OD with size of 800×800 for each image following [19], and each ROI is further resized to 256 × 256 and normalized by min-max normalization. The

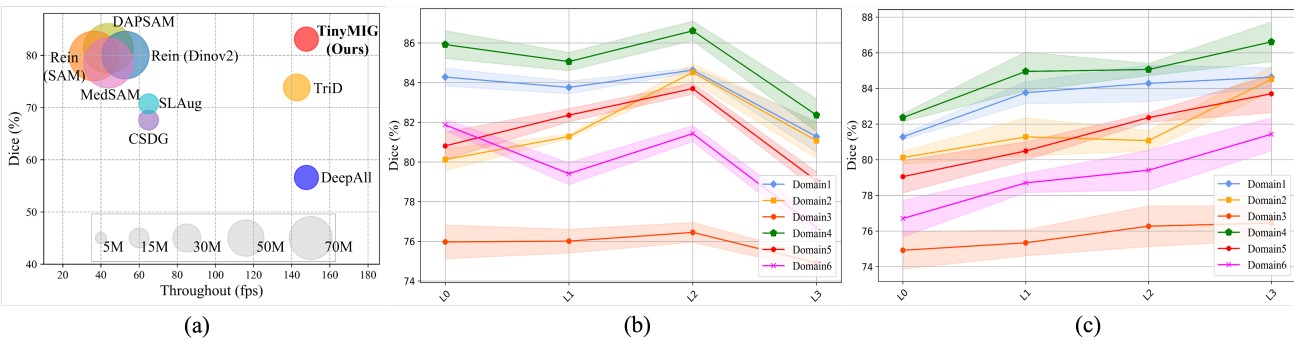

*Figure 8.* (a) The proposed TinyMIG significantly surpasses the SDG capabilities of SOTAs with extremely low computational cost and model parameter size. (b) Parameter analysis on the location of UFDM. (c) Parameter analysis on the location of LRA.

Dice Similarity Coefficient (DSC) and Average Surface Distance (ASD) are utilized for evaluation in this task.

**The Prostate segmentation task** comprises 116 MRI instances from six different clinical centers, aggregated from three public datasets, including NCI-ISBI13 (Bloch et al., 2015), I2CVB (Lemaître et al., 2015), and PROMISE12 (Litjens et al., 2014) datasets. Following the methodologies described in (Liu et al., 2020a; 2022a), the dataset is preprocessed to standardize the field of view for the prostate region and resized to $384 \times 384$ in the axial plane. The assessment of prostate segmentation performance also employs the Dice Similarity Coefficient (DSC).

### B.2. Experimental Details

We employ the AdamW optimizer (Loshchilov & Hutter, 2018) on all three medical image segmentation tasks, with $\beta = [0.9, 0.999]$. The initial learning rates are set as $l_0 = 0.0001$. These rates decay according to the polynomial rule $l_t = l_0 \times (1 - \frac{t}{T})^{0.9}$, where $l_t$ denotes the learning rate at epoch $t$, and $T$ represents the total number of epochs, which are set to 200 for prostate segmentation and 100 for the joint segmentation of OD/OC and Polyp segmentation, with the batch size set as 8. All our experiments are conducted on two 4090 GPU computing servers. Based on the experiments, we set $[\lambda_1, \lambda_2, \lambda_3, \lambda_4, \lambda_5]$ as [1,1,0.5,0.5,0.5] empirically. In addtion, we select the 3-6-9-12 layers in foundation models and 0-1-2-3 layers in the specialized model, i.e., $\mathcal{I}_R = (3, 6, 9, 12), \mathcal{I}_S = (0, 1, 2, 3)$ in Eq. 12.

## C. Extended Experiments.

**Segmentation Performance and Inference Efficiency of TinyMIG on Unseen Domain Samples**. As shown in Fig. 8 (a), thanks to the efficiency of the TinyMIG framework, our method achieves significant advantages in both segmentation performance (Dice) and inference speed (fps) during the testing phase. The horizontal axis represents inference speed, and the vertical axis represents segmentation performance on unseen domain samples in the prostate task. Compared to the state-of-the-art methods DPSAM, our TinyMIG not only achieves superior segmentation performance on unseen domain samples but also offers significantly better inference efficiency: fewer model parameters (22.52M vs. 98.45M), faster inference speed (147.74 vs. 43.91).

**Location of LRA.** Fig. 8 (c) examines the impact of inserting the LRA module at different network layers on performance. Similar to the positions investigated for UFDM, we evaluate the effect of varying insertion points. The results show that skipping any layer results in a slight performance drop. This demonstrates that the LRA module effectively filters domain-specific features and enhances domain-invariant features, enabling its integration at all layers to improve the capability of generalizable representation transferring of the VFM models.

**Location of UFDM.** We investigated the impact of UFDM insertion at different positions, as shown in Fig. 8 (b). L0 represents UFDM insertion after the first Conv-BN-ReLU layer (Layer 0), while L1 and L3 correspond to insertion after the first two and all ResNet layers, respectively. L1 yields the best performance, with a slight decline in L0 and L2. However, we observed a significant performance drop when UFDM was inserted at L3. This may be due to the fact that shallow layers in deep neural networks capture more style-related channel statistics, whereas deeper layers tend to encode more semantic information.

*Table 9.* Experiments on the Fundus dataset *w.r.t* different loss terms.

| $\lambda_1$ | $\lambda_2$ | $\lambda_3$ | $\lambda_4$ | $\lambda_5$ | Fundus Dataset |
|---|---|---|---|---|---|
| 1.0 | 1.0 | 1.0 | 1.0 | 1.0 | 83.18 |
| 1.0 | 1.0 | 1.0 | 1.0 | 0.5 | 83.66 |
| 1.0 | 1.0 | 1.0 | 0.5 | 1.0 | 83.72 |
| 1.0 | 1.0 | 0.5 | 1.0 | 1.0 | 83.29 |
| 1.0 | 1.0 | 1.0 | 1.0 | 2.0 | 82.96 |
| 1.0 | 1.0 | 1.0 | 2.0 | 1.0 | 82.30 |
| 1.0 | 1.0 | 2.0 | 1.0 | 1.0 | 82.37 |
| 1.0 | 0.5 | 0.5 | 0.5 | 0.5 | 84.12 |
| 0.5 | 1.0 | 0.5 | 0.5 | 0.5 | 84.27 |
| 1.0 | 1.0 | 0.5 | 0.5 | 0.5 | **84.97** |

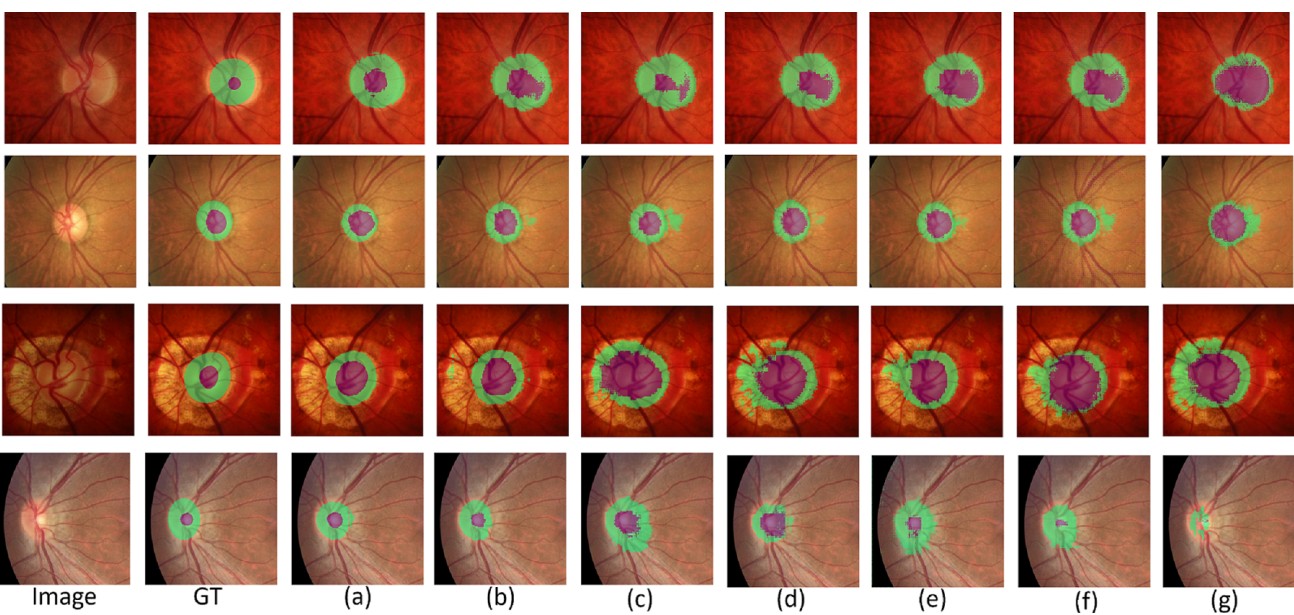

Image   GT   (a)   (b)   (c)   (d)   (e)   (f)   (g)

*Figure 9.* Comparisons across different SDG Methods on fundus imaging with ground truth (GT) and predictions (a-g). The subfigures (a) to (g) correspond to: (a) *TinyMIG*, (b) DAPSAM (Wei et al., 2024b), (c) TriD (Chen et al., 2023), (d) VPTTA (Chen et al., 2024), (e) MAD (Qu et al., 2023), (f) SLAug (Su et al., 2022b), (g) baseline.

**The impact of hyperparameters for loss terms.** We also present the results for detailed weight settings for loss terms to Tab. 9. Based on these results, our TinyMIG method is not particularly sensitive to the hyperparameters for loss terms. In this work, we select a set of hyperparameters $[\lambda_1, \lambda_2, \lambda_3, \lambda_4, \lambda_5] = [1.0, 1.0, 0.5, 0.5, 0.5]$ that exhibit the best performance as the default settings.

**More Qualitative results.** As shown in Fig. 9 and Fig. 10, we present additional visual results on the fundus and prostate tasks. Visual comparisons reveal striking advantages: TinyMIG precisely captures anatomical boundaries in unseen domains, whereas competing methods exhibit tissue confusion and boundary leakage artifacts, particularly in low-contrast regions. This demonstrates our method's superior generalization to distribution shifts.

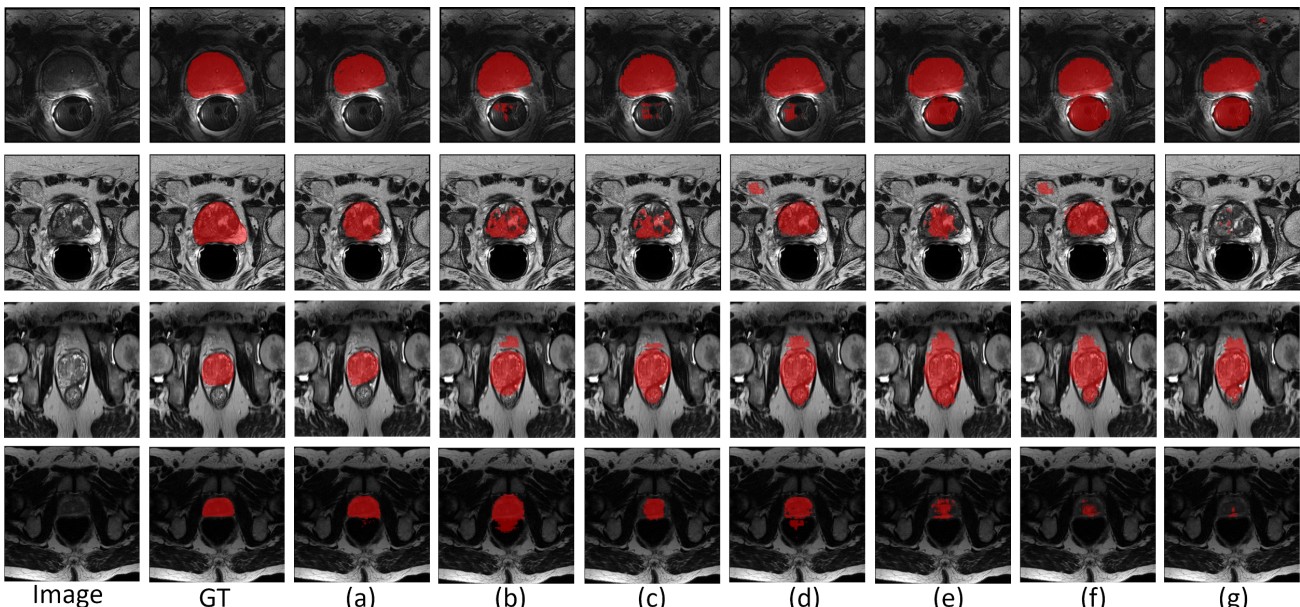

| Image | GT | (a) | (b) | (c) | (d) | (e) | (f) | (g) |

*Figure 10.* Comparisons across different SDG Methods on prostate imaging with ground truth (GT) and predictions (a-g). The subfigures (a) to (g) correspond to: (a) *TinyMIG*, (b) DAPSAM (Wei et al., 2024b), (c) TriD (Chen et al., 2023), (d) VPTTA (Chen et al., 2024), (e) MAD (Qu et al., 2023), (f) SLAug (Su et al., 2022b), (g) baseline.

