# OpenReview forum: "TinyMIG: Transferring Generalization from Vision Foundation Models to Single-Domain Medical Imaging"
_ICML.cc/2025/Conference — ICML 2025 poster_

### Official Review · Reviewer_peNL · 2025-03-13

**Overall Recommendation:** 4

**Summary:**

This paper addresses the challenge of single-source domain generalization in medical imaging by proposing a novel framework that transfers generalization knowledge from large vision foundation models to specialized smaller models, thereby eliminating the inference burden of large models during the testing phase.
Specifically, the authors introduce a dual-level knowledge transfer approach, encompassing both global and local perspectives. The proposed method demonstrates superior segmentation performance and inference efficiency across two medical domain generalization tasks.

**Claims And Evidence:**

1.	The UFDM module requires a more detailed explanation, as Figure 3a is somewhat ambiguous in conveying its functionality.
2.	The operation mixing the statistical feature parameters of the vision foundation model with Gaussian noise, as opposed to directly substituting the specialized model's feature parameters, needs further elaboration.

**Essential References Not Discussed:**

Frequency based domain generalization has been widely studied in image processing and machine learning. In the related work, please discuss the novelties with some strongly related works, such as:

[a] "Learning generalized segmentation for foggy-scenes by bi-directional wavelet guidance." Proceedings of the AAAI Conference on Artificial Intelligence. Vol. 38. No. 2. 2024.

[b] "Learning spectral-decomposited tokens for domain generalized semantic segmentation." Proceedings of the 32nd ACM International Conference on Multimedia. 2024.

[c] "Learning Frequency-Adapted Vision Foundation Model for Domain Generalized Semantic Segmentation." Annual Conference on Neural Information Processing Systems, 2024.

**Experimental Designs Or Analyses:**

1 The hyperparameter a of the beta distribution in the UFDM module is set to 0.1 without experimental justification.

2 On which layers is UFDM applied, and why these specific layers?

3 On which layers of both the specialized and foundation models is LRA applied, and what is the rationale for selecting these layers?

**Methods And Evaluation Criteria:**

Yes, the proposed methods and evaluation criteria are well-suited for the problem and application at hand. The usage of benchmark datasets, such as fundus and prostate imaging datasets, is appropriate for evaluating the method's segmentation performance and generalization capabilities. Additionally, the evaluation metrics, including inference speed and parameter count, are relevant for assessing the practicality and efficiency of the proposed approach in real-world medical scenarios.

**Other Comments Or Suggestions:**

No

**Other Strengths And Weaknesses:**

- Strengths

1.	The paper introduces an innovative knowledge distillation framework designed to transfer generalization knowledge from large vision foundation models to specialized segmentation models. This approach is both novel and practically significant, as it enhances the efficiency of model deployment in medical applications.

2.	The authors propose a dual-level distillation approach, which includes learning the global feature distribution priors from the large model and distilling domain-invariant features from each layer. This comprehensive strategy significantly improves the learning effectiveness of the smaller model.

3.	The method achieves impressive segmentation results on two medical imaging tasks, along with a notable improvement in inference efficiency, making it highly suitable for real-world clinical deployment.

4.	The paper provides extensive ablation experiments, including detailed quantitative analyses, model parameter comparisons, and inference efficiency evaluations. These thorough experiments offer valuable insights into the method's robustness and effectiveness.

- Weaknesses

1.	The UFDM module requires a more detailed explanation, as Figure 3a is somewhat ambiguous in conveying its functionality.

2.	The rationale behind mixing the statistical feature parameters of the vision foundation model with Gaussian noise, as opposed to directly substituting the specialized model's feature parameters, needs further elaboration.

3.	The hyperparameter a of the beta distribution in the UFDM module is set to 0.1 without experimental justification.

**Questions For Authors:**

1.	On which layers is UFDM applied, and why these specific layers?

2.	On which layers of both the specialized and foundation models is LRA applied, and what is the rationale for selecting these layers?

3.	The hyperparameter a of the beta distribution in the UFDM module is set to 0.1 without experimental justification.

**Relation To Broader Scientific Literature:**

This paper contributes to the broader scientific literature by bridging the gap between large-scale vision foundation models and domain-specific medical imaging tasks. The proposed TinyMIG framework is inspired by recent advancements in foundation models, such as GPT and SAM, which have shown remarkable generalization abilities. However, unlike prior work that primarily focuses on direct fine-tuning or adaptation, this paper introduces a novel knowledge distillation strategy that emphasizes both global feature distribution consistency and local fine-grained alignment.

This approach is reminiscent of techniques like AdaIN for style transfer and frequency-domain methods for domain adaptation, but it uniquely combines these ideas to address the specific challenges of medical imaging.

**Theoretical Claims:**

I have reviewed the theoretical claims related to the UFDM module and the spectral transformation properties. Specifically, I examined the statistical parameter transformation within the UFDM module and its justification in the context of domain generalization. The idea of integrating Gaussian noise with statistical feature parameters  is well-founded and aligns with existing literature on robust feature learning. Additionally, the decomposition and filtering of phase and amplitude components in the spectral domain are reasonable, as phase information is known to capture structural details while amplitude relates to global characteristics. These theoretical underpinnings are consistent with established principles in signal processing and domain adaptation, supporting the validity of the proposed approach.

---

> ### Author Rebuttal · Authors · 2025-04-01
>
> Dear Reviewer peNL:
>
> **1. discuss the novelties with some strongly related works**
>
> We sincerely appreciate your valuable feedback. Below, we clarify the key distinctions and advantages of our method compared to the cited works from two perspectives: motivation and implementation. These points will be thoroughly addressed in our revised manuscript.
>
> *(1) Motivation:* The cited methods essentially perform fine-tuning by directly adjusting intermediate features of large foundation models in the frequency domain. In contrast, our approach transfers generalizable knowledge from foundation models to specialized models via frequency-domain transformation. Crucially, our method eliminates the need for foundation model inference during testing, making it significantly more efficient than fine-tuning-based approaches.
>
> *(2) Implementation:* Our frequency-domain content enhancement module requires only two lightweight filters to achieve generalized feature transfer. This design is notably simpler than existing solutions, which rely on additional parametric network layers or computationally intensive attention mechanisms.
>
> **2. The Issue of UFDM module**
>
> We sincerely appreciate your constructive suggestions. We are pleased to provide further clarification about the UFDM module and will incorporate these details in our revised manuscript. Below, we elaborate on both the implementation and motivation:
>
> (1) Implementation:
> The UFDM (Universal Feature Distribution Modulation) employs Adaptive Instance Normalization (ADAIN) to perturb the primary channel statistics of the specialized model by leveraging the style priors from large foundation models. This design enables effective style transfer while maintaining the model's core functionality.
>
> (2) Motivation:
> Vision foundation models acquire comprehensive image representations from large-scale natural image datasets, which inherently capture diverse stylistic variations during training. By exploiting these rich style priors through our UFDM module, we significantly enhance the generalization capacity of oriented object detectors when encountering novel domains with previously unseen styles.
>
> (3) Experimental Validation:
> As demonstrated in Figure 5, our UFDM module substantially outperforms existing feature perturbation approaches across all evaluation metrics. These compelling results validate the superiority and effectiveness of our proposed method.
>
> **3. The rationale behind mixing the statistical feature**
>
> Building upon this framework, we innovatively adapt the mixup technique from image augmentation to effectively blend the channel-wise prior distributions between the specialized model and the vision foundation model. This strategic implementation serves two key purposes:
>
> (1) It generates samples with significantly enhanced stylistic diversity, and
>
> (2)Through our proposed consistent bidirectional KL divergence loss, it robustly facilitates the specialized model's ability to learn domain-invariant features across these diversified style variations.
>
> **4.Experimental Analysis of UFDM/LRA Module Location and Hyperparameter α**
>
> (1) We sincerely apologize for placing the experimental results regarding UFDM and LRA module positioning in the supplementary materials (Figure 8(b)-(c)). The findings reveal several important insights: Figure 8(b) demonstrates that bypassing any network layer leads to measurable performance degradation, confirming that the LRA module successfully suppresses domain-specific features while effectively preserving domain-invariant characteristics. Furthermore, Figure 8(c) indicates optimal performance is achieved at the L1 position, with marginally reduced effectiveness at L0 and L2. Notably, we observed substantial performance deterioration when implementing UFDM at the L3 level. This phenomenon likely stems from the fundamental property of deep neural networks where shallow layers predominantly capture style-related channel statistics, while deeper layers primarily encode high-level semantic information.
>
> (2) We sincerely apologize for any confusion regarding the ablation studies of parameter α. Our selection of α=0.1 was informed by the established MixStyle methodology, a canonical approach for statistical parameter mixup. The parameter α serves as the shape parameter of the beta distribution governing the convex weight λ in Equation 2, where smaller α values skew λ toward extreme values (0 or 1), while larger values concentrate λ around 0.5.
>
> We conducted ablation studies on prostate dataset as shown in the table below, systematically evaluated α values across the range of 0.1 to 0.3. The results demonstrate minimal performance variance across these parameter settings, with α=0.1 emerging as the most stable configuration, consistent with the optimal parameters identified in prior MixStyle implementations.
>
> |α|0.1(**Ours**)|0.1|0.3|
> |:-:|:-:|:-:|:-:|
> |**Dice $↑$**|83.09|82.69|82.39|
> |**ASD $↓$**|3.06|3.31|3.49|

---

> > ### Comment · Reviewer_peNL · 2025-04-05
> >
> > The authors have adequately addressed my previous concerns, particularly with the additional experimental data which helps support their claims. Given these improvements, I recommend increasing the score of this work by one point. Thank you for your detailed responses.

---

> > > ### Author Response · Authors · 2025-04-07
> > >
> > > We sincerely appreciate your recognition of our work and your valuable suggestions. We are also deeply grateful for the time and effort you have dedicated to reviewing our manuscript.

---

### Official Review · Reviewer_3XUJ · 2025-03-13

**Overall Recommendation:** 4

**Summary:**

This paper proposes to address the challenge of single-source domain generalization in the field of medical imaging. The authors propose to leverage the generalization knowledge embedded in current vision foundation models to enhance the generalization capabilities of smaller-capacity segmentation models. They enables the smaller segmentation models to learn generalization knowledge through both global and local learning during the training phase, thereby eliminating the need for additional inference with the vision foundation models. Experiments are conducted on two datasets to validate the effectiveness.

**Claims And Evidence:**

1. The rationale behind introducing a phase component attention mechanism after the learnable filters in the local alignment module needs further elaboration to better understand its contribution to domain-invariant feature enhancement.
2. Experiments with varying sizes of small models, such as a ResNet-18-based UNet, are needed to fully demonstrate the generalizability of the TinyMIG framework.

**Essential References Not Discussed:**

None

**Experimental Designs Or Analyses:**

* Can the Localized Representation Alignment Loss be replaced with other forms of constraint functions?
* In the FDCE module, is the learnable filter for the phase component necessary? To the best of my knowledge, phase components are generally stable. Why is there a need to filter "domain-varying features" in this context?

**Methods And Evaluation Criteria:**

The methods and evaluation criteria are aligned with the problem of single-source domain generalization in medical imaging. The framework effectively leverages knowledge distillation from foundation models, and the usage of benchmark datasets such as fundus and prostate imaging provides a solid basis for evaluating segmentation performance and generalization capabilities.

**Other Comments Or Suggestions:**

None

**Other Strengths And Weaknesses:**

# Strengths
* The paper is well-structured, with clear
* It achieves good results on fundus and prostate datasets, improving segmentation performance while maintaining low inference speed and parameter count.
* The release of code helps in this field, promoting broader adoption and research.

# Weaknesses
* The integration of Gaussian noise with statistical feature parameters in the UFDM module, rather than directly replace the smaller model's parameters, requires a more detailed explanation to elucidate its role in enhancing generalization capabilities.
* The local alignment module's phase component attention mechanism, applied after two learnable filters, needs a clearer justification regarding its interaction with domain-invariant features and its overall contribution to the module's functionality.
* While the role of different foundation models within TinyMIG has been explored, additional experiments with varying sizes of small models, such as a ResNet-18-based UNet, are also necessary.

**Questions For Authors:**

The rationale behind introducing a phase component attention mechanism after the learnable filters in the local alignment module needs further elaboration.

**Relation To Broader Scientific Literature:**

This approach is reminiscent of techniques like AdaIN for style transfer and frequency-domain methods for domain adaptation, but it uniquely combines these ideas to address the specific challenges of medical imaging.

**Theoretical Claims:**

The use of AdaIN-based methods to generate mixed statistical parameters is widely recognized and accepted within the domain generalization community, lending credibility to the proposed approach.

---

> ### Author Rebuttal · Authors · 2025-04-01
>
> Dear Reviewer 3XUJ,
>
> We sincerely appreciate your comprehensive review and highly constructive feedback on our manuscript. Below we provide point-by-point responses to address all of your concerns:
>
> 1. **phase component attention mechanism**
> We sincerely appreciate your feedback. In the field of domain generalization, the magnitude spectrum and phase spectrum are typically regarded as representations of style and content information. Since the phase spectrum exhibits more stable and generalizable content characteristics, we employ phase-spectrum-based attention to align and enhance content features between the foundation model and the specialized model. This strategy further mitigates interference from domain-specific style information, thereby facilitating more effective transfer of generalizable features.
>
> 2. **more experiments about vary sizes of small models**
> We sincerely appreciate your valuable suggestion regarding model improvement. As recommended, we have conducted additional experiments with different specialized model sizes (ResUNet-18) to further validate our approach. The comprehensive results are presented in Table [X] below, which demonstrates consistent performance across varying model capacities.
>
> |Task|**S-model**|#Param|Fundus Seg. (DSC)|Prostate Seg. (DSC)|
> |:-:|:-:|:-:|:-:|:-:|
> |Prostate|Unet(ResNet-18)| 11.93 M | 79.26| 81.83|
> |Prostate|Unet(ResNet-34)| 22.54 M | 84.97| 83.09|
>
> 3. **Localized Representation Alignment Loss**
> We sincerely appreciate your insightful comment. As demonstrated in Tab. 7 of our comparative experiments examining different LRA implementations, our proposed domain-invariant feature alignment method consistently outperforms other knowledge distillation approaches. This performance advantage primarily derives from our spectral dynamic filtering mechanism, which achieves two critical objectives: (1) enabling effective cross-domain generalizable feature transfer while (2) actively suppressing domain-specific feature interference within the specialized model.
>
>
> 4. **Necessity of the FDCE-P Module**
>
> We thank the reviewer for their careful evaluation.
>
> (a) The necessity of the FDCE-P Module is motivated by several key observations: In visual domain generalization, learning domain-invariant representations from frequency features is crucial. Prior works confirm that the phase component encodes content features (domain-invariant), while the amplitude component captures style features (domain-specific). Accordingly, we propose dual spectral filters to separately process these components, precisely eliminating style artifacts while preserving content features. This design significantly enhances the transfer of generalizable representations from vision foundation models.
>
> (b) Furthermore, Our quantitative results in Table 6 empirically demonstrate the necessity of the learnable phase component filter (FDCE-P). The synergistic operation of both magnitude and phase spectral filters enables effective transfer of generalization capabilities from visual foundation models to specialized models.
>
> 5. **The noise mixup of UFDM**
>
> We sincerely appreciate your insightful suggestions. We adopt random mixup between foundation model features and noise for two reasons:
>
> (1) Vision foundation models learn image representations from large-scale natural image datasets, which encompass diverse styles during training. This rich stylistic variability enhances the generalization capability of specialized model when applied to unseen target domains with novel styles.
>
> (2) Mixup (vs. full features) enhances the stylistic diversity of perturbed features by simultaneously incorporating partial style priors from the foundation model and randomized noise components, thereby facilitating the specialized model’s learning of domain-invariant features. The introduction of random noise compels the model to maintain semantic consistency across a broader spectrum of stylistic variations, effectively promoting the imitation of the base model’s globally generalized features.

---

### Official Review · Reviewer_uvPv · 2025-03-13

**Overall Recommendation:** 3

**Summary:**

The authors propose a transfer generalization framework, TinyMIG, to address the issue of single-domain generalization caused by the diversity of imaging devices and variabilities among data collection centers. To capture both global feature distribution and local fine-grained details, they developed a global distribution consistency learning strategy and a localized representation alignment method to mimic the prior distribution of the foundation model, promoting semantic alignment and generalization distillation. They validated their model on two domain generalization segmentation benchmarks.

**Claims And Evidence:**

Yes, the claims made in the submission are supported by clear and convincing evidence.

**Essential References Not Discussed:**

The authors covered and discussed SOTA literature related to the domain.

**Experimental Designs Or Analyses:**

I reviewed the complete experimental design and analysis section, including the performance comparison of TinyMIG with SOTAs, ablation studies, and t-SNE/Heatmap visualizations.

**Methods And Evaluation Criteria:**

Yes, the proposed methods and evaluation criteria are well-suited to the problem, with relevant benchmark datasets and effective metrics for assessing the network's performance.

**Other Comments Or Suggestions:**

The current rating reflects the generalizability of their proposed approach with any vision foundation model, solid experimentation, and ablation studies, paving the way for improved domain generalization tasks compared to SOTA models. I strongly recommend the authors to consider the raised weaknesses and concerns and address them in the rebuttal.

**Other Strengths And Weaknesses:**

**Strengths**

*S1.* The paper is well-written, with clear pictorial depictions and a solid methodological development.

*S2.* The integration of instance normalization (AdaIN) within the global distribution consistency learning module is unique.

*S3.* The authors conducted extensive ablation studies and cross-validation, validating the effectiveness of various network parameters, loss functions, and hyperparameters.

*S4.* The problem is well-motivated, and the experiments are comprehensive, covering two domain generalization benchmarks and comparing their performance with SOTA models.


**Weaknesses**

*W1. Concern about domain-invariant features.* I am unsure how minimizing the bidirectional KL divergence between the probabilistic distributions of the semantic predictions would encourage the predictor to capture domain-invariant features. While I agree that this strategy would better align the predictions across two domains, the underlying feature representations might still contain domain-specific biases that are not addressed by the loss in Eq. (4). Furthermore, KL divergence does not directly ensure domain-invariant features, as it only aligns soft predictions (according to the equations), not the underlying representations. Additionally, the model could overfit to the modulated distribution, and relying on soft predictions alone may not capture meaningful domain invariance, leaving domain-specific biases unaddressed in the feature representations.

*W2. Motivation for frequency-based alignment.* I found the purpose and motivation for developing an alignment loss based on frequency-based discriminative content enhancement unclear. How do they ensure improved robustness in unseen domains?

*W3. Experimentation on complex anatomical segmentation datasets.* While the current experimentation is comprehensive, I would like to see additional experiments on brain anatomical segmentations using diverse datasets (e.g., OASIS, ABIDE, ADNI, etc.). This would further strengthen the validity of the proposed approach and its applicability to a wider range of medical imaging data.*

**Minor Concerns.**

- FDCA module/L_LRA is missing in the figure, although it is mentioned in the caption.
- The feature reprogramming block needs further explanation and thorough validation. How is it effective in improving network. performance?

**Questions For Authors:**

Please see the weaknesses.

**Relation To Broader Scientific Literature:**

The contributions are very much related to the broader scientific literature, can be integrated with any vision foundation models, in terms of medical image segmentation tasks.

**Theoretical Claims:**

I believe the paper is more focused on practical applications.

---

> ### Author Rebuttal · Authors · 2025-04-01
>
> Dear Reviewer uvPv,
>
> **R-W1 KL for Domain-invariant Features**
>
> We sincerely appreciate your insightful concerns. The bidirectional KL divergence serves two key purposes in our approach:
>
> (1) It ensures stable and consistent semantic outputs before and after feature perturbation. This is because the perturbed features share identical semantic content with the original features while exhibiting different styles. Thus, the bidirectional KL divergence emphasizes invariant pixel-level semantic information across styles, enhancing model stability and reducing sensitivity to style variations.
>
> (2) As you rightly pointed out, global semantic alignment alone is insufficient.
> Our additional local representation alignment directly matches intermediate-layer features at a fine-grained level, further facilitating domain-invariant feature learning. These two components work synergistically to enable the generalization transferring from foundation model to specialized model.
>
> Also, we are glad to refine the description regarding 'KL-divergence-based domain-invariant feature learning' to adopt a more measured tone.
>
> **R-W2 Motivation for Frequency-Domain Alignment and how it works**
>
> *(1) Motivation for Frequency-Domain Alignment:*
>
> Our approach is motivated by several key observations:
>
> (a) In visual domain generalization, a critical research direction involves learning domain-invariant content representations from frequency-space features. Prior studies [r1,r2] have well-established that phase components predominantly encode domain-invariant content, while amplitude components capture domain-specific style properties.
>
> (b) As shown in Appendix Fig. 7, foundation models exhibit more generalized features, demonstrating stable responses across diverse domains. We thus leverage their intermediate features as teacher guidance to optimize our amplitude and phase filters more effectively, enhancing the student model’s imitation of domain-invariant representations from the foundation model.
>
> [r1] Wang, Kunyu, et al. "Generalized uav object detection via frequency domain disentanglement." Proceedings of the IEEE/CVF conference on computer vision and pattern recognition. 2023.
>
> [r2] Xu, Qinwei, et al. "A fourier-based framework for domain generalization." Proceedings of the IEEE/CVF conference on computer vision and pattern recognition. 2021.
>
> *(2) How to ensure the robustness:*
>
> (a) We first decouple spatial features into amplitude spectra (style-related) and phase spectra (content-related) via Fourier transform. An amplitude filter suppresses domain-specific amplitude variations, while a phase filter reinforces domain-invariant phase features. Both filters are optimized via Eq. (13)’s local alignment loss, encouraging the student model to mimic the teacher’s generalized frequency-domain characteristics.
>
> (b) As evidenced in Table 4, our Frequency-Domain Consistency Enhancement (FDCE) significantly improves generalization capcity. Moreover, the heatmap visualizations in Fig. 6 (columns 2 vs. 3) demonstrate that without Localized Representation Alignment (LRA), the model’s attention is distracted by complex and stylic backgrounds, whereas FDCE enables precise focus on target organs in unseen domains.
>
> **R-W3 More experiments on brain anatomical datasets**
>
> We sincerely appreciate your valuable suggestion regarding expanding the experimental datasets.
> We have already made several efforts in this direction:
>
> (1) Comprehensive Metrics: To thoroughly validate our method's effectiveness, we employed three evaluation metrics - DSC, ASD, and HD (as detailed in our response to Reviewer #1).
>
> (2) Statistical Rigor: Beyond average metrics, we additionally incorporated standard deviation measurements to assess our method's performance stability.
>
> (3) In direct response to your insightful suggestion, we have now supplemented our evaluation with an additional polyp segmentation benchmark [r3] (We sincerely apologize that this differs from your suggested brain dataset - we will carefully consider your recommendation for future work). This new evaluation consists of four public datasets from different medical centers:D1 (CVC-ClinicDB): 380 images, D2 (CVC-ColonDB): 612 images, D3 (ETIS): 196 images and D4 (Kvasir): 1000 images.
> As shown in the following table, our method achieves sota performance on this challenging benchmark, further demonstrating its effectiveness. We are deeply grateful for your thoughtful comments and will incorporate all these improvements in our revised manuscript.
>
> |Methods|ERM|MixStyle|CSDG|MaxStyle|EFDM|SLAug|TriD|MAD|VPTTA|DAPSAM|SAMed|**Ours**|
> |:-:|:-:|:-:|:-:|:-:|:-:|:-:|:-:|:-:|:-:|:-:|:-:|:-:|
> |DSC↑|54.82|66.50|67.60|67.60|62.62|68.01|67.20|71.17|72.83|74.74|71.93|**77.92**|
>
> [r3] Chen, Ziyang, et al. "Each test image deserves a specific prompt: Continual test-time adaptation for 2d medical image segmentation." Proceedings of the IEEE/CVF conference on computer vision and pattern recognition. 2024.

---

> > ### Comment · Reviewer_uvPv · 2025-04-06
> >
> > The authors have adequately addressed some of my earlier concerns. I will maintain my original score. However, the paper would be further strengthened by experiments on complex brain anatomical segmentation using diverse datasets (e.g., OASIS, ABIDE, ADNI), which would help validate the method’s broader applicability in medical imaging.

---

> > > ### Author Response · Authors · 2025-04-08
> > >
> > > Thank you very much for your recognition of our work and your sincere suggestions. We are also deeply grateful for the time and effort you have devoted to reviewing the manuscript, which has helped us further improve its quality. Once again, we sincerely appreciate your support!

---

### Official Review · Reviewer_mFeP · 2025-03-14

**Overall Recommendation:** 3

**Summary:**

The authors propose a method to efficiently make use of generalizable features of visual foundation models by distilling learned distributions to a smaller, more efficient model. Thereby, they propose four main components:

1. Global Distribution Consistency Learning: Forcing the “specialized model” (i.e. student model) to be less sensitive towards changes in distribution by augmenting the feature statistics (i.e. mean and sd) of the student across different layers with the statistics of the foundation models layers.
2. Cross-Distribution Consistency Loss: Forcing the segmentations of the student with and without the augmented features to be closer.
3. Localized Representation Alignment: Forcing the features of the transformed foundation models features and the student’s features to be close.
4. Frequency-based Discriminative Content Enhancement: Enforcing phase and amplitude of the FFT transformed student and teacher features to be close. Thereby preserving semantics (phase) and increasing robustness towards style shifts (amplitude)

The authors show superiority of their methods compared to a variety of different methods.

**Claims And Evidence:**

The evidence is mostly clear and convincing with the exception for line 368 to 371, where the authors refer to a result that cannot be seen in the figure. Furthermore, there are no results for the ASD for the OC/OD segmentation task. Finally, Table 7 and 8 seem like the results are not carefully pasted, as the first two columns of both tables are the same to the second digit.

**Essential References Not Discussed:**

The authors included all citations necessary to understand their work and according to my knowledge are not missing any important references.

**Experimental Designs Or Analyses:**

The proposed modules and corresponding losses seem to be sensible and mathematically sound, except for a typo in γmix where is should be σs  and μs instead of σ(s) and μ(s).

One of the vital parts of the architecture design, the “FRM” block, is not described. Instead, the authors refer to a non-existing section. Furthermore, the authors refer to the appendix for further information regarding the network architecture of the specialized model, which also cannot be found in the appendix. This makes the paper non-reproducible and opaque.

Despite claiming to use Dice and ASD for evaluation of both datasets, the authors fail to provide the ASD results for the OD/OC segmentation task.

The authors claim superiority of their UFDM compared to other style randomization-based DG methods and thereby refering to figure 3. However, no such evidence can be seen in figure 3 (which is a description of the method) nor in any other figure in the paper.

**Methods And Evaluation Criteria:**

The authors compare their method against several relevant domain generalization methods- However, comparing their method against the base architecture of their “specialized model” as well as a fine-tuned version of the used foundational model is desirable to clearly see the effect of their method. (In some figures, the authors use a “baseline” method, but since there is no reference to the method in the paper, it is unclear what is being shown there.)

The datasets chosen in the paper are appropriate for providing evidence for the authors claims. However, a more detailed overview of the prostate dataset is desirable (i.e. number of images per center)

The chosen metrics to compare the different methods are appropriate, however, the authors should consider adding Hausdorff distance for completeness.

In addition to the mean segmentation metrics, standard deviations would be also helpful to assess the significance of differences in mean.

**Other Comments Or Suggestions:**

None

**Other Strengths And Weaknesses:**

The paper lacks clarity and reproducibility (e.g. missing description of baseline, unclear method “ERM” in tables, missing description of a vital part of the architecture (FRA)).

Furthermore, in the paper are several mistakes such as missing contend that was referred to, multiple references that are two times in the reference sections, wrong order of the legends in Figure 5, allegedly wrongly pasted results in the first two columns of Table 7/8.

**Questions For Authors:**

None

**Relation To Broader Scientific Literature:**

The paper is in line with previous findings that foundational models generalize well [1]. The authors show that the specialized model performs well despite being less complex than the teacher (foundation) model. This is coherent with literature as well. [2] The authors combine methods from the literature with their own ideas. (e.g. DAPSAM [3], Squeeze and Excitation [4] and  AdaIN [5] with their FDCE module.)

[1] Oquab, M., Darcet, T., Moutakanni, T., et al. (2024). DINOv2: Learning Robust Visual Features without Supervision. arXiv preprint arXiv:2304.07193. https://arxiv.org/abs/2304.07193.

[2] Hinton, G., Vinyals, O., & Dean, J. (2015). Distilling the Knowledge in a Neural Network. arXiv preprint arXiv:1503.02531. https://arxiv.org/abs/1503.02531.

[3] Wei, Z., Dong, W., Zhou, P., et al. (2024). Prompting Segment Anything Model with Domain-Adaptive Prototype for Generalizable Medical Image Segmentation. arXiv preprint arXiv:2409.12522. https://arxiv.org/abs/2409.12522.

[4] Hu, J., Shen, L., Albanie, S., et al. (2019). Squeeze-and-Excitation Networks. arXiv preprint arXiv:1709.01507. https://arxiv.org/abs/1709.01507.

[5] Huang, X., & Belongie, S. (2017). Arbitrary Style Transfer in Real-time with Adaptive Instance Normalization. arXiv preprint arXiv:1703.06868. https://arxiv.org/abs/1703.06868.

**Theoretical Claims:**

There are no theoretical proofs in the paper.

---

> ### Author Rebuttal · Authors · 2025-04-01
>
> Dear Reviewer mFeP,
> We sincerely appreciate your comprehensive review and highly constructive feedback on our manuscript. Below we provide point-by-point responses to address all of your concerns:
>
> **1.Some Typos**:
>
> *(1) Revised: Comparative Results of Perturbation Methods vs. UFDM (Lines 368-371)*:
>
> We sincerely apologize for this oversight. The comparative analysis of perturbation methods was indeed detailed in Table 5 and Section 4.3. We have now updated the implementations of DSU and MaxStyle as shown in the revised table.
>
> ||Mixstyle|CrossNorm|TriD|DSU|Maxstyle|**Ours**|
> |:-:|:-:|:-:|:-:|:-:|:-:|:-:|
> |Dice↑| 81.34| 81.78| 82.03|81.47|81.86| **83.09**|
> |ASD↓|3.52|3.48|3.27|3.50|3.34|**3.06**|
>
> Notably, UFDM achieves superior performance by leveraging the style prior of foundation models. We will carefully finalize Table 5 and correct the citation error (Lines 368-371).
>
> *(2) The ASD of OC/OD:*
>
> We sincerely apologize for this oversight. The experiments were indeed conducted but inadvertently omitted from the appendix. The experimental results are shown below:
>
> |Metric|ERM|MixStyle|CSDG|MaxStyle|EFDM|DSU|SLAug|TriD|MAD|VPTTA|DAPSAM|SAMed|**Ours**|
> |:-:|:-:|:-:|:-:|:-:|:-:|:-:|:-:|:-:|:-:|:-:|:-:|:-:|:-:|
> |**OD Avg↓**|12.90|11.75|11.52|11.16|10.24|10.33|10.46|9.84|9.46|9.46|12.75|10.68|**5.66**|
> |**OC Avg↓**|16.59|13.09|12.58|11.92|10.76|11.28|11.36|10.29|10.28|10.28|13.48|11.58|**6.03**|
>
> (3) We correct this lapse. The corrected data for Table 8 is provided below. As shown, only the MSE loss significantly impacts the results, while other consistency losses show limited effects.
>
> |Methods|MSE|KL-div|JS-div|**Ours**|
> |:-:|:-:|:-:|:-:|:-:|
> |Dice↑|81.21|82.73|82.47|**83.09**|
> |ASD↓|3.79|3.15|3.23|**3.06**|
>
> **2.The Description of Baseline, FRM and Specialized Models**:
>
> (1) **Baseline**: We sincerely apologize for any confusion caused. As detailed in Section four (Lines 244-245), we adopt ResUnet34 as the specialized model for fair comparison with other SDG methods. For the baseline, we adopted a ResNet34-UNet model without additional domain generalization components.
> We will clarify this further to improve readability and more willing to make all the code fully open-source to facilitate reproducibility and contribute to the advancement of the community.
>
> (2) **FRM**: We sincerely appreciate your thorough review. As noted in Lines 269-271, the FRM's primary role is to transform foundation model features through a simple convolutional block (conv-BN-ReLU) for channel/resolution alignment with the pyramidal features of specialized model (Unet with a Resnet34 backbone), enabling Localized Representation Alignment. While essential, this component is not our core technical contribution, which explains its brief treatment. We are willing to expand the technical details in revision and open-source all code to ensure reproducibility.
>
> **3. Additional Dataset Descriptions and Evaluation Metrics**:
>
> (1) **Prostate dataset**: The prostate dataset details are provided below, consistent with prior SDG works (e.g., DAPSAM, CSDG). Post identical preprocessing, the slice counts per domain are tabulated. We sincerely appreciate your feedback. Additional details can be found in Appendix B.1.
> ||D1|D2|D3|D4|D5|D6|
> |:-:|:-:|:-:|:-:|:-:|:-:|:-:|
> |Cases|30|30|19|13|12|12|
> |Slices|381|354|449|162|249|145|
>
> (2) **HD metric**: We sincerely appreciate your suggestion regarding the Hausdorff Distance (HD), a critical metric for segmentation evaluation. As shown in the comparative results below, our method achieves the best HD performance on the prostate dataset, demonstrating its effectiveness. We would be delighted to include this evaluation metric in our revisions.
>
> |Method|ERM|MixStyle|CSDG|MaxStyle|EFDM|DSU|SLAug|TriD|MAD|VPTTA|DAPSAM|SAMed|**Ours**|
> |:-:|:-:|:-:|:-:|:-:|:-:|:-:|:-:|:-:|:-:|:-:|:-:|:-:|:-:|
> |HD↓|73.51|65.74|47.22|54.91|53.71|48.31|45.91|45.72|46.12|41.06|35.76|43.89|**30.03**|
>
> (3) **standard deviations**: Yes, we are very happy to include standard deviations on prostate dataset to ensure a fairer comparison of all methods, as shown in the table below. We would be pleased to incorporate this addition in the revised manuscript. We sincerely appreciate your suggestion.
> |Method|ERM|MixStyle|CSDG|MaxStyle|EFDM|DSU|SLAug|TriD|MAD|VPTTA|DAPSAM|SAMed|**Ours**|
> |:-:|:-:|:-:|:-:|:-:|:-:|:-:|:-:|:-:|:-:|:-:|:-:|:-:|:-:|
> |DSC↑|56.59±8.79|57.06±6.32|70.37±5.61|67.58±5.63|68.27±5.91|69.37±3.96|70.71±4.22|71.74±6.59|69.37±2.89|74.58±3.77|81.31±1.28|78.51±4.36|**83.09±0.57**|
>
> I sincerely apologize for any minor errors caused by my limited writing skills, which may have inconvenienced your review. If anything remains unclear, please do not hesitate to point it out—I will carefully consider and refine it. Once again, I truly appreciate your valuable suggestions for improving the quality of the paper!
> **Finally, We will implement these modifications in the revision and open-source all the code/weights for reproducibility.**

---

### Decision · Program_Chairs · 2025-05-01

**Decision:**

Accept (poster)

**Comment:**

The paper received positive reviews unianimously from four experts. The rebuttal further boosts the final score. Accept.